# Intermediate filament network perturbation in the *C. elegans* intestine causes systemic dysfunctions

Florian Geisler[1], Sanne Remmelzwaal[2], Vera Jankowski[3], Ruben Schmidt[2], Mike Boxem[2], Rudolf E Leube[1]*

[1]Institute of Molecular and Cellular Anatomy, RWTH Aachen University, Aachen, Germany; [2]Division of Developmental Biology, Institute of Biodynamics and Biocomplexity, Department of Biology, Faculty of Science, Utrecht University, Utrecht, Netherlands; [3]Institute for Molecular Cardiovascular Research, University Hospital RWTH Aachen, Aachen, Germany

**\*For correspondence:**
rleube@ukaachen.de

**Competing interest:** The authors declare that no competing interests exist.

**Abstract** Intermediate filaments (IFs) are major components of the metazoan cytoskeleton. A long-standing debate concerns the question whether IF network organization only reflects or also determines cell and tissue function. Using *Caenorhabditis elegans*, we have recently described mutants of the mitogen-activated protein kinase (MAPK) SMA-5 which perturb the organization of the intestinal IF cytoskeleton resulting in luminal widening and cytoplasmic invaginations. Besides these structural phenotypes, systemic dysfunctions were also observed. We now identify the IF polypeptide IFB-2 as a highly efficient suppressor of both the structural and functional deficiencies of mutant *sma-5* animals by removing the aberrant IF network. Mechanistically, perturbed IF network morphogenesis is linked to hyperphosphorylation of multiple sites throughout the entire IFB-2 molecule. The rescuing capability is IF isotype-specific and not restricted to *sma-5* mutants but extends to mutants that disrupt the function of the cytoskeletal linker IFO-1 and the IF-associated protein BBLN-1. The findings provide strong evidence for adverse consequences of the deranged IF networks with implications for diseases that are characterized by altered IF network organization.

## Editor's evaluation

This is a very detailed and rigorous study using the power of worm genetics and phenotypic characterization to investigate defects in intermediate filament assembly and organization and its effects on tissue mechanobiology, particularly in the intestine. The work also has implications for understanding disease pathologies.

## Introduction

Intermediate filaments (IFs) together with actin filaments and microtubules are important components of the cytoskeleton. They mediate mechanical tissue stability and have been implicated in multiple cellular processes such as vesicle transport, organelle positioning, cell cycle regulation, differentiation, metabolism, motility, and stress response (*Etienne-Manneville, 2018*; *Jacob et al., 2018*; *Leube and Schwarz, 2016*; *Margiotta and Bucci, 2016*; *Yoon and Leube, 2019*; *Coch et al., 2020*; *Toivola et al., 2010*; *Geisler and Leube, 2016*). An ongoing debate is whether IFs are simply bystanders of these cellular processes or contribute actively to cell function and dysfunction (*Etienne-Manneville, 2018*; *Bott and Winckler, 2020*; *Ridge et al., 2022*).

To study the morphogenesis and function of the IF system, we use the nematode *Caenorhabditis elegans*. A striking example of specialized IF network organization is encountered in the intestine, where six IF polypeptides IFB-2, IFC-1, IFC-2, IFD-1, IFD-2, and IFP-1 co-localize in the apical cytoplasm and form the electron dense endotube, which surrounds the lumen as a compact fibrous sheath and is attached to the composite *C. elegans* apical junction (CeAJ) (*Carberry et al., 2009*; *Jahnel et al., 2016*; *Bossinger et al., 2004*). It is assumed that this evolutionary conserved localization of the intestinal IF network (*Coch and Leube, 2016*) provides protection against mechanical stress (*Geisler and Leube, 2016*; *Toivola et al., 2010*; *Geisler et al., 2016*). Accordingly, the endotube is positioned at the interface between the cortical actin cytoskeleton with the stiff microvillar brush border and the soft cytoplasm (*Geisler et al., 2020*; *McGhee, 2007*; *Bossinger et al., 2004*). Because of its high degree of elasticity the IF-rich endotube likely dampens mechanical stresses occurring during food intake, defecation, and body movement (*Geisler et al., 2020*). Elimination of the intestinal IFs IFB-2, IFC-2, and IFD-2 therefore leads to luminal widening although loss of IFC-1, IFD-1, and IFP-1 does not (*Geisler et al., 2020*). Electron microscopy revealed that IFC-2 mutants have a rarefied endotube, whereas IFB-2 mutants lack it completely (*Geisler et al., 2019*; *Geisler et al., 2020*). The intestinal IF mutants present very mild organismal phenotypes with only minor or no detectable effects on development, progeny production, survival, and stress sensitivity with the exception of IFC-2 mutants, which cause additional excretory canal defects (*Geisler et al., 2020*; *Al-Hashimi et al., 2018*; *Carberry et al., 2009*; *Karabinos et al., 2001*; *Karabinos et al., 2004*).

Modulators of intestinal IF distribution have been identified and characterized in *C. elegans* (*Carberry et al., 2012*; *Geisler et al., 2016*; *Stutz et al., 2015*; *Estes et al., 2011*; *Geisler et al., 2019*; *Remmelzwaal et al., 2021*; *Koyuncu et al., 2021*). One of them is SMA-5, a stress-activated kinase orthologue of the mitogen-activated protein kinase (MAPK) type (*Watanabe et al., 2005*). The *C. elegans* WormBase (version WS287; https://wormbase.org) lists MAPK7 (BMK1; ERK4; ERK5; PRKM7) as its closest orthologue in vertebrates. However, MAPK7 activity has not been linked to IFs so far. Abundant cytoplasmic invaginations of the adluminal, apical plasma membrane develop over time in loss-of-function *sma-5(n678)* mutant intestines (*Geisler et al., 2016*). These changes correlate with the development of a locally thickened endotube consisting of amorphous material next to areas with complete endotube loss. The cytoplasmic invaginations form at the transition between both areas. The structural changes go along with biochemical changes, that is, altered IFB-2 phosphorylation. Furthermore, in comparison to the wild type and to intestinal IF mutants *sma-5* mutants are smaller, produce less offspring, develop more slowly, live shorter, and are more sensitive to microbial pathogens and osmotic as well as oxidative stress (*Geisler et al., 2016*; *Geisler et al., 2019*). It is not known whether these pathologies are attributable to the altered IF cytoskeleton or other *sma-5(n678)*-dependent cellular perturbations. Another modulator of the intestinal IF cytoskeleton is the intestinal IF organizer gene *ifo-1* (*Carberry et al., 2012*), which was originally identified as a cellular defense gene against pore-forming toxins and as part of the MAPK/JNK innate immune defense network (referred to as *ttm-4* in *Kao et al., 2011*). The IF network in loss-of-function *ifo-1* mutants collapses into large aggregates, which accumulate primarily at the CeAJ and occasionally in the cytoplasm. *ifo-1* mutants are small, have reduced progeny, and are hypersensitive to different types of stress. The phenotypes are not only much more pronounced than those observed in IF mutants but are also more pronounced than in *sma-5* mutants (*Geisler et al., 2019*; *Geisler et al., 2020*; *Carberry et al., 2012*). Again, the contribution of the deranged IF network to the *ifo-1* mutant phenotype has not been determined to date. Recently, we described a third type of intestinal IF regulator, namely BBLN-1 (*Remmelzwaal et al., 2021*). BBLN-1 is a small coiled-coil protein whose depletion results in an intestinal phenotype that is highly reminiscent of that observed in *sma-5* mutants presenting bubble-shaped cytoplasmic invaginations of the apical plasma membrane. Systemic consequences of *bbln-1* mutation have not been analyzed in detail to date. Remarkably, all three regulators have been localized to the apical submembrane compartment in intestinal cells (*Remmelzwaal et al., 2021*; *Geisler et al., 2016*; *Carberry et al., 2012*).

To identify components of the IF regulatory pathways in the *C. elegans* intestine, we performed a suppressor screen of *sma-5(n678)* mutants, which identified *ifb-2* mutation as the most efficient suppressor. Loss-of-function *ifb-2* mutation also partially rescued the *ifo-1* and *bbln-1* phenotypes. The perturbed intestinal IF networks were completely depleted in all instances. Even more importantly, the systemic dysfunctions were also rescued for the most part. These findings provide new evidence on

the gain-of-toxic-function hypothesis driving the pathogenesis of aggregate-forming diseases, which have been shown to involve aberrant IF networks (e.g., *Coulombe et al., 2009*; *Chamcheu et al., 2011*; *Yoshida and Nakagawa, 2012*; *Clemen et al., 2013*; *Gentil et al., 2015*; *Didonna and Opal, 2019*).

## Results

### The loss-of-function phenotype of the MAP-kinase SMA-5 is rescued by depletion of the intestinal IF polypeptide IFB-2

To identify downstream targets of the MAP-kinase orthologue SMA-5, we performed a genome-wide suppressor screen in OLB18 carrying the loss-of-function allele *sma-5(n678)*. Selecting for fast developing lines we isolated two lines, in which we were not able to detect any native IFB-2 expression by immunoblotting (data not shown). We therefore focused on the *ifb-2* gene locus for further analysis and identified an 83 bp deletion (positions 5.751.812–5.751.893; allele *ifb-2(kc20)*). Due to the resulting frameshift the mutant *ifb-2* gene encodes only 120 amino acids, consisting of the entire 31 amino acid-long head domain, 85 amino acids of the coiled-coil rod domains 1a and 1b (*Karabinos et al., 2004*), and the four additional carboxyterminal amino acids R-G-P-S. To demonstrate that loss-of-function of IFB-2 is by itself sufficient and necessary to rescue the *sma-5(n678)* phenotype and to exclude that the residual aminoterminal part of IFB-2 that is still produced from *kc20* is responsible for the rescue, we crossed *sma-5(n678)* mutants with the previously described *ifb-2* knockout allele *kc14* (*Geisler et al., 2019*). The *kc14* allele encodes a 29 amino acid-long oligopeptide encompassing only 13 of the most aminoterminal amino acids of IFB-2. Assessment of body length revealed a near normal body size of the double *sma-5(n678);ifb-2(kc14)* mutants. Only a minor reduction was detected at day 4 after egg laying as was the case for *ifb-2(kc14)* single mutants (*Figure 1—figure supplement 1*). Normal body length was reached by day 5 in *sma-5(n678);ifb-2(kc14)* and *ifb-2(kc14)* but not in *sma-5(n678)* (not shown).

Light microscopy revealed a rescue of the luminal widening and cytoplasmic invagination phenotypes (*Figure 1A–D'*). Dextran feeding experiments confirmed these observations (*Figure 1—figure supplement 2*). Ultrastructural analyses further showed a reversion of the phenotype to a near wild-type situation in the double mutants (*Figure 1E–H'*). Only minor luminal undulations and slight perturbation of the microvillar brush border remained. Most notably, the endotube was completely absent (*Figure 1H–H'*). The ultrastructural features were identical to those encountered in *ifb-2(kc14)* single mutants (*Figure 1G–G'*).

The *ifb-2(kc14)* knockout allele furthermore restored the prolonged time of development of *sma-5(n678)* (*Figure 2A*). While still elevated in comparison to the wild type, developmental time was only slightly different from the single *ifb-2(kc14)* mutant. To define minor alterations in development more precisely, analysis of the different developmental stages was carried out next. The results shown in *Figure 2B* illustrate the high degree of similarity in the developmental time course of the single *ifb-2(kc14)* and double *sma-5(n678);ifb-2(kc14)* mutants. Both developed more slowly than the wild type and much faster than the *sma-5(n678)* animals, 7.1% of which never reached adulthood. The latter could be ascribed to larval arrest at the L4 stage (*Figure 2C*).

Life span determinations further showed that the shortened median life span of *sma-5(n678)* animals could be rescued in the *sma-5(n678);ifb-2(kc14)* double mutants to the level observed in *ifb-2(kc14)* single mutants. Yet, the life spans of *sma-5(n678);ifb-2(kc14)* and *ifb-2(kc14)* mutants were still reduced in comparison to the wild type by 1 day (*Figure 2D*). Similarly, the drastic reduction in progeny of *sma-5(n678)* was rescued in the double mutant to the level observed in *ifb-2(kc14)*, which, again, was significantly less than in the wild type (*Figure 2E*).

In a next set of experiments, we investigated whether loss of IFB-2 would also rescue the increased stress sensitivity of *sma-5(n678)* (*Geisler et al., 2019*). In the presence of oxidative stress, the survival of *sma-5(n678)* was reduced by 3.5 hr in comparison to wild-type N2 but was only reduced by 0.5 hr and 1.5 hr in *ifb-2(kc14)* and *sma-5(n678);ifb-2(kc14)*, respectively (*Figure 2F*). All mutants appeared to be similarly affected in osmotic stress assays and a statistically significant rescue phenotype could not be observed (*Figure 2G*).

Taken together, we can conclude that loss of IFB-2 rescues all major *sma-5(n678)* phenotypes, except osmotic stress hypersensitivity, to levels observed in *ifb-2(kc14)*. This demonstrates that intestinal IF network perturbations adversely affect organismal physiology.

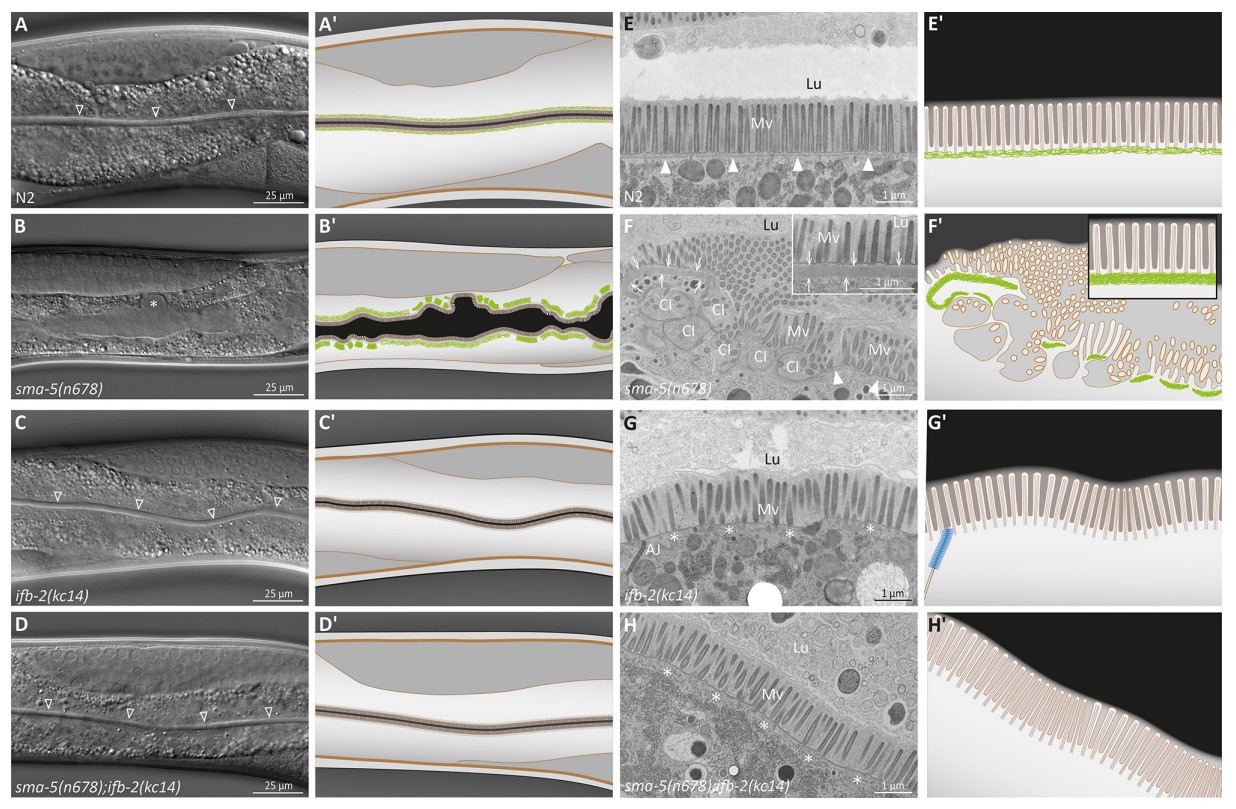

**Figure 1.** The *ifb-2(kc14)* knockout allele rescues the luminal widening and cytoplasmic invagination phenotypes and abolishes the aberrant intermediate filament (IF) network of *sma-5(n678)* mutants. (A–D') Differential interference contrast pictures and corresponding schematics of viable wild-type N2 (A, A'), *sma-5(n678)* (B, B'), *ifb-2(kc14)* (C, C'), and *sma-5(n678);ifb-2(kc14)* animals (D, D'). *ifb-2* knockout causes only minor intestinal defects. In contrast, *sma-5(n678)* animals display extensive luminal widening and large cytoplasmic invaginations of the apical plasma membrane in intestinal cells (asterisk in B). This phenotype is effectively reversed by the *ifb-2(kc14)* knockout allele. Non-filled arrowheads: normal-appearing intestinal lumen. (E–H') Electron microscopy images of high-pressure frozen samples and corresponding schematics show wild-type N2 (E', E'), *sma-5(n678)* (F, F'), *ifb-2(kc14)* (G, G'), and *sma-5(n678);ifb-2(kc14)* intestinal cell apices (H, H'). *sma-5(n678)* animals contain regions with an enlarged endotube consisting of densely packed IFs (arrows) and differently sized cytoplasmic invaginations (CI) with no endotube or a reduced endotube (arrowheads). Additional knockout of *ifb-2* almost restores the wild-type morphology with only residual luminal widening and mildly perturbed microvillar arrangement. But the endotube, which is easily detected in the wild type (arrowheads), is completely absent in single *ifb-2(kc14)* and double *sma-5(n678);ifb-2(kc14)* mutants (expected position marked by asterisks). Lu, lumen; Mv, microvilli; AJ, *C. elegans* apical junction.

The online version of this article includes the following figure supplement(s) for figure 1:

**Figure supplement 1.** Depletion of IFB-2 rescues growth defects in *sma-5(n678)*, *ifo-1(kc2)*, and *bbln-1(mib70)* loss-of-function mutants.

**Figure supplement 2.** Depletion of IFB-2 rescues the luminal dilation phenotype in *sma-5(n678)*, *ifo-1(kc2)*, and *bbln-1(mib70)* loss-of-function mutants.

## Depletion of individual intestinal IF polypeptides reveals isotype-specific rescue efficiency of the *sma-5(n678)* phenotype

To test whether the observed rescue is specific for IFB-2 or applies also to the other five IFs that are expressed in the intestine, each IF was downregulated by RNAi in the *sma-5(n678)* background. As simple readout, F1 progeny from RNAi-treated worms was imaged on agar plates 4 days after egg laying (*Figure 3A–G*) and the body length was measured (*Figure 3H*). The assays confirmed the expected efficient rescue of the developmental growth defect in *sma-5(n678)* by *ifb-2(RNAi)*. An easily detectable, though reduced rescuing efficiency was detected for *ifc-2(RNAi)*. An even less pronounced but statistically still significant rescue could be identified in *ifd-1*, *ifd-2*, and *ifp-1* RNAi-treated animals and none for *ifc-1(RNAi)*.

We next examined whether the cytoplasmic invagination phenotype could be rescued in a similar fashion. To this end, the different IF-encoding RNAs were downregulated in the IFB-2a::CFP reporter strain (*Figure 3I–O'*). As predicted, *ifb-2(RNAi)* abolished the reporter fluorescence as well as the cytoplasmic invaginations of *sma-5(n678)*. *ifc-2(RNAi)* also led to a reduction of the cytoplasmic

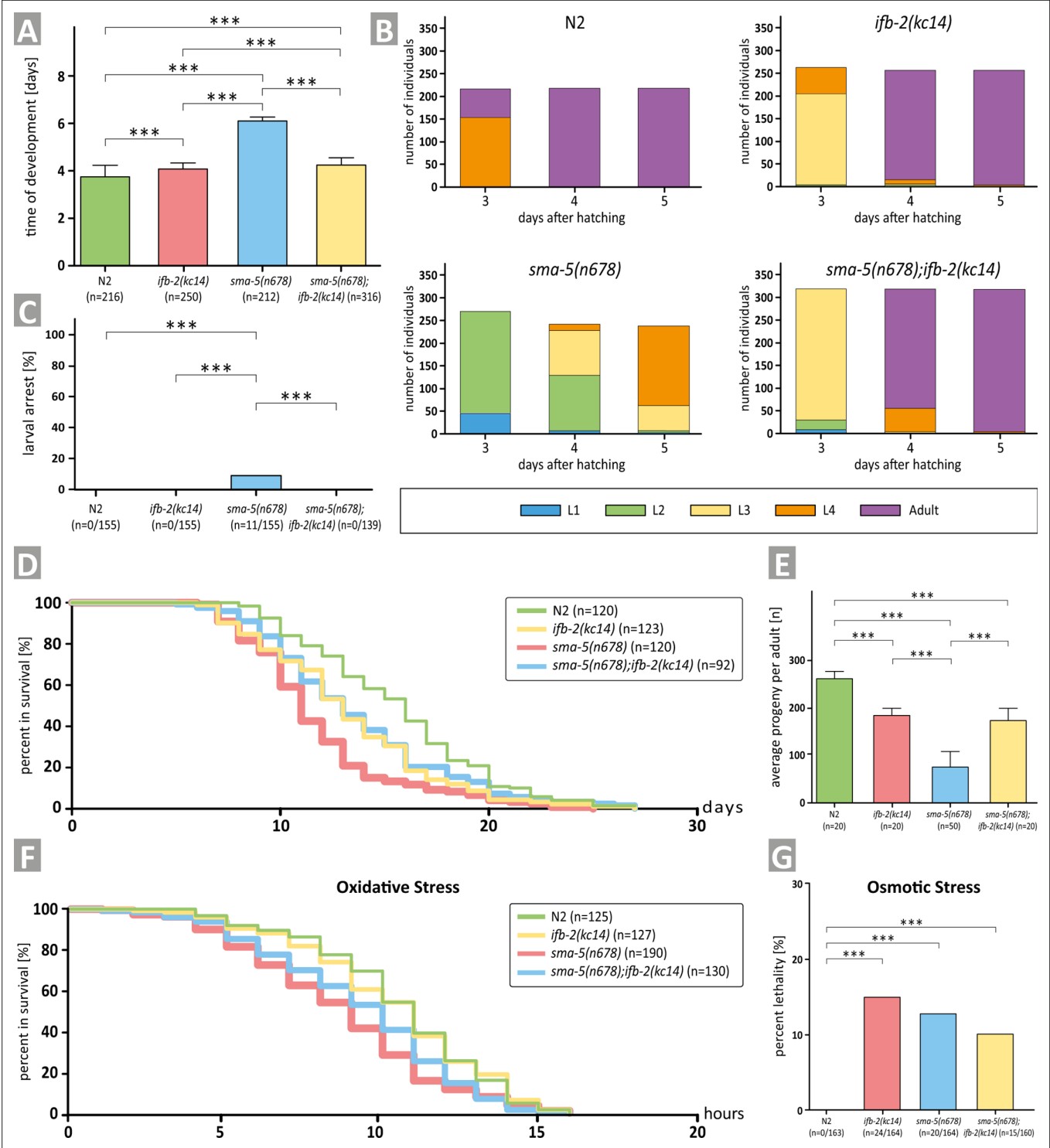

**Figure 2.** Depletion of IFB-2 (*ifb-2(kc14)*) rescues developmental retardation, larval arrest, reduced median life span, decreased brood size, and increased sensitivity to oxidative stress of *sma-5(n678)* mutants. (**A**) The histogram shows a comparison of the time of development in N2, *ifb-2(kc14)*, *sma-5(n678)*, and *sma-5(n678);ifb-2(kc14)* (N2: 3.7±0.5 days; *ifb-2(kc14)*: 4.1±0.2 days; *sma-5(n678)*: 6.0±0.2 days; *sma-5(n678);ifb-2(kc14)*: 4.2±0.4 days; ***p<0.0001). (**B**) The color-coded histograms depict the number of larval and adult stages detected 3, 4, and 5 days after hatching. (**C**) The histogram illustrates a complete rescue of the larval arrest phenotype observed in *sma-5(n678)* by *ifb-2(kc14)* (N2: 0%; *ifb-2(kc14)*: 0%; *sma-5(n678)*: 7.1%; *sma-5(n678);ifb-2(kc14)*: 0%). (**D**) The plot shows that the reduced life span of *sma-5(n678)* is rescued by addition of *ifb-2(kc14)* to the level encountered in *ifb-2(kc14)* but not the wild-type level (median survival for N2: 16 days; *ifb-2(kc14)*: 13 days; *sma-5(n678)*: 11 days; *sma-5(n678);ifb-2(kc14)*: 13 days; p=0.0004 *ifb-2(kc14)* versus N2; p<0.0001 *sma-5(n678)* versus N2; p=0.0003 *sma-5(n678)* versus *ifb-2(kc14)*; p<0.0001 *sma-5(n678);ifb-2(kc14)* versus N2;

*Figure 2 continued on next page*

*Figure 2 continued*

p<0.0046 *sma-5(n678);ifb-2(kc14)* versus *sma-5(n678)*). (**E**) The histogram reveals that the drastic reduction in progeny observed in *sma-5(n678)* mutants versus N2 (76±31 vs 263±13; p<0.0001) is rescued in *sma-5(n678);ifb-2(kc14)* double mutants (175±21 vs 76±31; p<0.001) but does not reach wild-type level (175±21 vs 263±13; p<0.001) and is similar to *ifb-2(kc14)* (175±21 vs 183±15; p>0.5). (**F**) The survival plot shows the effect of acute oxidative stress in the wild type (N2), *ifb-2(kc14)*, *sma-5(n678)* and *sma-5(n678);ifb-2(kc14)* backgrounds (median survival for N2: 11 hr; *ifb-2(kc14)*: 11 hr; *sma-5(n678)*: 9 hr; *sma-5(n678);ifb-2(kc14)*: 10 hr; p<0.0001 for N2 or *ifb-2(kc14)* versus *sma-5(n678)*; p<0.05 for *sma-5(n678);ifb-2(kc14)* versus *sma-5(n678)*; p<0.01 for N2 or *ifb-2(kc14)* versus *sma-5(n678);ifb-2(kc14)*). (**G**) The histogram scores the percentage of dead worms in response to acute osmotic stress for N2 (0%), *ifb-2(kc14)* (14.6%), *sma-5(n678)* (12.2%), and *sma-5(n678);ifb-2(kc14)* (9.4%).

invagination phenotype but was much less efficient than *ifb-2(RNAi)*. All other interfering RNAs, however, did not visibly affect the invagination phenotype. While we cannot exclude that the presence of the reporter obscured the effect of RNA downregulation of these IF polypeptides, these results are overall consistent with the rescuing activity on body length.

To find out whether the observed rescue after *ifb-2(RNAi)* is caused by an overall loss of the intestinal IF network, we downregulated *ifb-2* expression in all six intestine-specific IF reporter strains (*Figure 3—figure supplement 1*). While IFB-2 was essential for IFC-2, IFD-1, and IFD-2 network formation, IFC-1 and IFP-1 were still able to form a network-like structure at the subapical domain, although only to a very limited extent. Note, however, that IFC-1 and IFP-1 fluorescence was not due to expression of the respective endogenous genes but due to fosmids, which may have resulted in overexpression.

Taken together, the isotype-specific phenotypes can be explained by the heteropolymerization of the different IF polypeptides and their abundance. Thus, IFB-2 has been shown to pair with multiple intestinal IFs (*Karabinos et al., 2017*) and has been identified as the main regulator of the endotube (*Geisler et al., 2020* and results of this study). It is therefore not surprising that IFB-2 depletion elicits the strongest rescuing activity of the *sma-5(n678)* mutant phenotype. It is also in line with IFC-2 depletion providing the second strongest rescuing activity, as also evidenced in previously reported effects on endotube structure and stress sensitivity (*Geisler et al., 2020*). Overall, our findings demonstrate that the altered IF network rather than loss of a single IF is responsible for the observed rescue of the *sma-5(n678)* phenotype.

## Aminoterminal IFB-2 domains contribute to intestinal IF network morphogenesis and function

In search for a molecular mechanism that may be involved in altered IF network morphogenesis, it is of interest that the head domains of mammalian IF polypeptides have been implicated in cytoplasmic IF assembly (*Omary et al., 2006*; *Zhou et al., 2021*; *Hatzfeld and Burba, 1994*). Given the essential contribution of IFB-2 to intestinal IF network formation, we concentrated on this polypeptide. We deleted base pairs 4–1107 of the *ifb-2* gene by CRISPR-cas resulting in a truncated protein missing the complete aminoterminal head domain (*ifb-2(mib169[S2-M43 deleted])II*) (cf. *Karabinos et al., 2004*). Mutants showed a phenotype reminiscent of that described for *sma-5(n678)* animals with cytoplasmic invaginations of the intestinal lumen accompanied by a rarefied IFC-2 fluorescence with regions completely lacking a signal next to areas showing enhanced fluorescence (*Figure 4A–B′*). These observations provide evidence for the evolutionary conserved importance of the IF head domain for IF network formation.

Previous observations had also shown that phosphorylation sites in the head domain of IFs are most frequently targeted by kinases with major consequences on IF network dynamics and organization (review in *Sawant and Leube, 2017*; *Snider and Omary, 2014*). In accordance, Wormbase and Phosida list multiple serine phosphorylation sites in the IFB-2 head domain. We therefore decided to mutate putative phosphorylation targets in the IFB-2 head domain by introducing phosphomimetic S>D and phosphorylation-deficient S>A mutations into all serines of the head region, namely S2, S5, S7, S16-S19, S24, and S35 (alleles *kc22* and *kc26*, respectively). Generating the mutant *ifb-2* alleles in the IFC-2a/e::YFP background allowed direct examination of the effect on intestinal IF network formation. Neither the phosphomimetic nor the phosphorylation-deficient mutant animals presented a visible change in IFC-2 localization (*Figure 4C–E*). Further characterization revealed that the mutants showed no developmental delay (*Figure 4G*) and only a slight reduction of progeny in case of the *ifb-2* head S>D mutants (*Figure 4H*). However, significant reduction of life span could be observed in

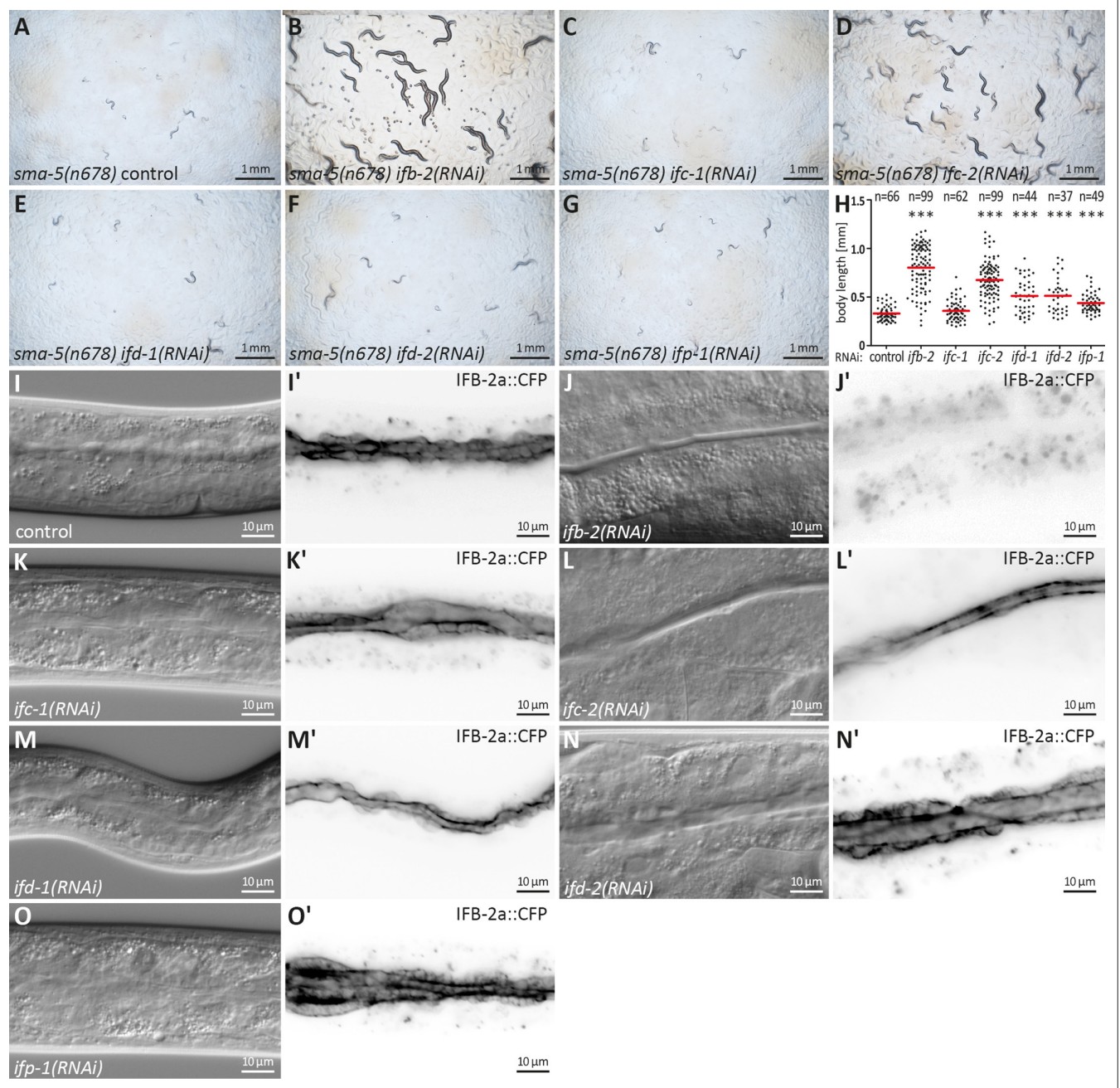

**Figure 3.** Downregulation of *ifb-2* and *ifc-2* is most efficient in suppressing developmental retardation, small body size, and cytoplasmic invaginations of *sma-5(n678)*. (**A–G**) show bright-field images of agar plates containing F1 *sma-5(n678)* subjected to RNAi 4 days after egg laying. (**H**) The scatter dot blot summarizes the results of body length measurements in *sma-5(n678)* subjected to either empty RNAi vector (control; 331±72.71 µm), *ifb-2(RNAi)* (802.90±235.10 µm), *ifc-1(RNAi)* (358.50±103.50 µm), *ifc-2(RNAi)* (675.60±194.10 µm), *ifd-1(RNAi)* (511.70±172.90 µm), *ifd-2(RNAi)* (512.60±178.80 µm) or *ifp-1(RNAi)* (443.30±109 µm). Note that the strongest rescue is observed for *ifb-2* followed by *ifc-2* and trailed by *ifd-1, ifd-2,* and *ifp-1* all of which are statistically significant (p<0.0001). No detectable rescue is observed for *ifc-1*. (**I–O'**) The microscopy images show differential interference contrast at left and corresponding fluorescence detection (inverse presentation) of the IFB-2a::CFP reporter in vital *sma-5(n678)* animals after RNAi against *ifb-2* (**J, J'**), *ifc-1* (**K, K'**), *ifc-2* (**L, L'**), *ifd-1* (**M, M'**), *ifd-2* (**N, N'**), and *ifp-1* (**O, O'**) (RNAi control in **I, I'**). Note that the knockdown of *ifb-2* leads to a loss of reporter fluorescence and efficiently rescues the *sma-5(n678)* invagination phenotype. A rescue is also detectable after loss of IFC-2 albeit at reduced efficiency. None of the other knockdowns resulted in detectable reduction of the invagination phenotype.

The online version of this article includes the following figure supplement(s) for figure 3:

**Figure supplement 1.** Loss of IFB-2 prevents network formation of IFC-2, IFD-1, and IFD-2, while IFC-1 and IFP-1 network formation is severely reduced.

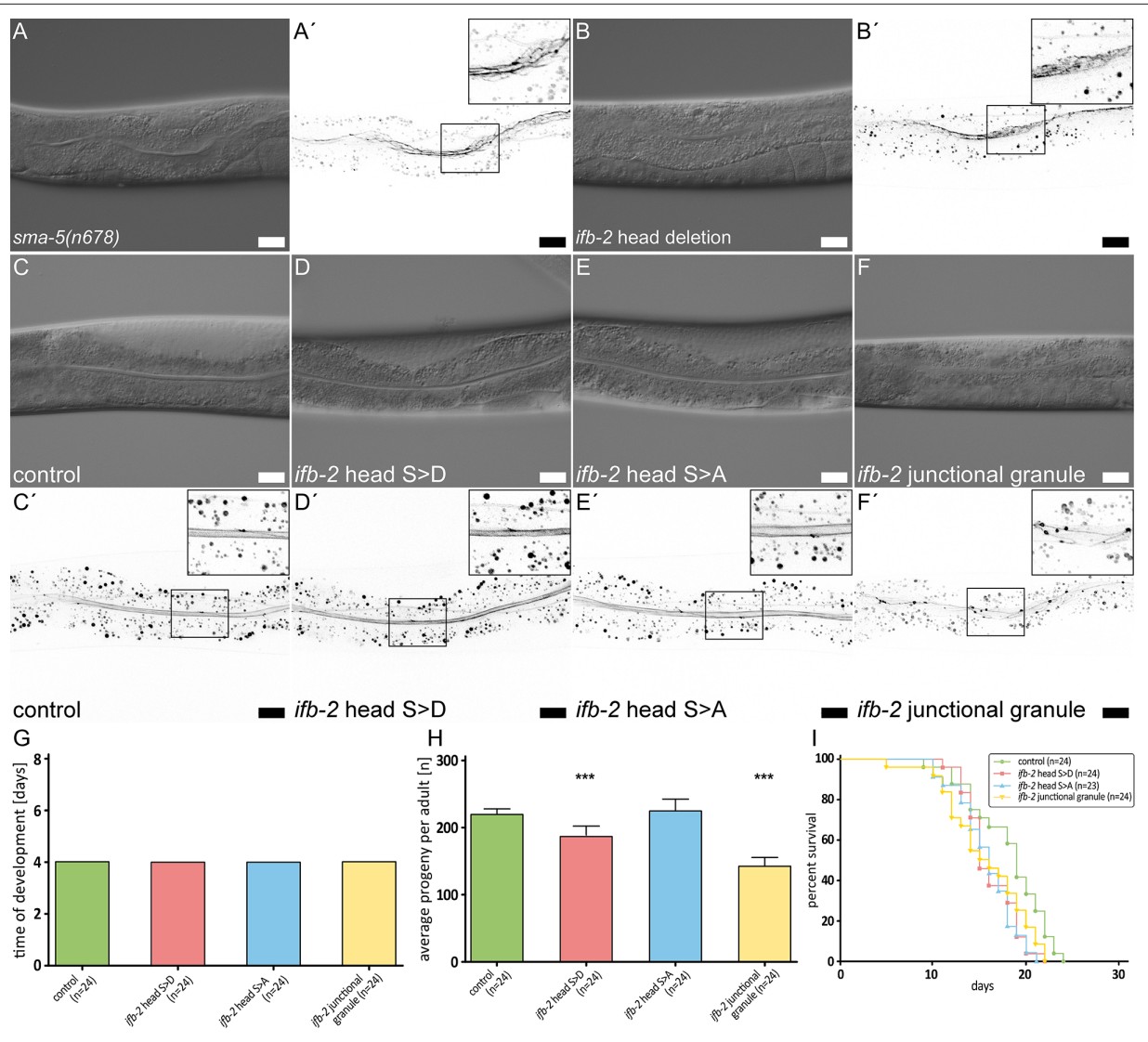

**Figure 4.** The IFB-2 aminoterminus is involved in intestinal intermediate filament (IF) network morphogenesis, progeny production, and life span. (**A–F'**) The microscopy images show differential interference contrasts (**A–F**) and corresponding fluorescence recordings of the IFC-2a/e::YFP reporter (inverse presentation; **A'–F'**) in vital *sma-5(n678)* (**A–A'**), *ifb-2* head deletion (**B–B'**, *ifb-2(mib169[D2-M43 deleted])II*), control (**C–C'**), phosphomimetic *ifb-2* head S>D (**D–D'**, *ifb-2(kc22[S2D;S5D;S7D;S16-19D;S24D;S35D])II*), phosphodeficient *ifb-2* head S>A (**E–E'**, *ifb-2(kc26[S2A;S5A;S7A;S16-19A;S24A;S35A]) II*) and 'junctional granule' mutants (**F–F'**, *ifb-2(kc27[S2A;S5A;S7A;S16-19A;E31-A184 deleted])II*). Scale bars: 20 μm. (**G–H**) The histograms present a comparison of time of development (**G**) and average progeny (**H**) of control, phosphomimetic *ifb-2* head S>D, phosphodeficient *ifb-2* head S>A and 'junctional granule' mutants. None of the mutants shows a prolonged development (control: 4.0 days; *ifb-2* head S>D: 4.0 days; *ifb-2* head S>A: 4.0 days; 'junctional granule' mutant: 4.0 days). The average progeny per adult, however, is reduced in *ifb-2* head S>D but not in *ifb-2* head S>A mutants in comparison to control (control: 227±9; *ifb-2* head S>D: 193±16; *ifb-2* head S>A: 232±18; p<0.0001 control versus *ifb-2* head S>D). Note the even higher reduction in progeny observed in 'junctional granule' mutants versus control (control: 227±9; junctional granule mutant: 145±16; p<0.0001). (**I**) The survival plot shows that mutants with genetic modifications of the *ifb-2* head domain have a reduced life expectancy albeit to different degrees with the *ifb-2* head S>D mutant showing the most severe phenotype (median survival for control: 19 days; *ifb-2* head S>D: 15 days; *ifb-2* head S>A: 16 days; junctional granule mutant: 15.5 days; p=0.0471 *ifb-2* head S>D versus control; p=0.0206 *ifb-2* head S>A versus control; p=0.0468 junctional granule mutant versus control; p<0.0001).

The online version of this article includes the following figure supplement(s) for figure 4:

**Figure supplement 1.** *ifb-2(kc27)* animals show IFC-2 positive junctional granules in concert with an overall reduced IFB-2/IFC-2-containing network.

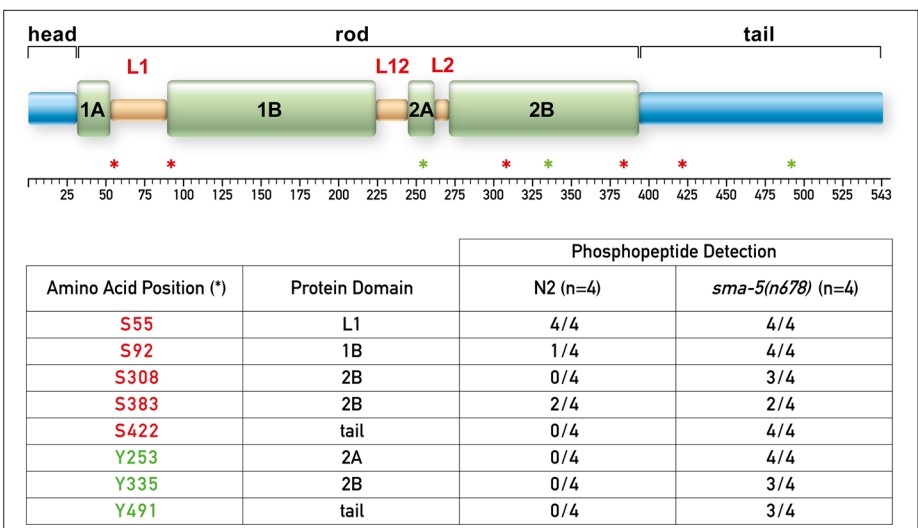

**Figure 5.** IFB-2 is hyperphosphorylated at multiple sites in the absence of SMA-5. The upper panel shows a domain model of IFB-2 with its aminoterminal head, alpha helical central rod (coil 1A, linker L1, coil 1B, linker L12, coil 2A, linker L2, and coil 2B), and carboxyterminal tail. Positions of phosphorylated amino acids are marked by asterisks (serines in red, tyrosines in green). The table below lists the number of samples, in which the respective phosphopeptides were identified, and their domain localization.

The online version of this article includes the following source data and figure supplement(s) for figure 5:

**Source data 1.** Identified phosphopeptides in *sma-5(n678)* and their corresponding domain localization.

**Figure supplement 1.** Examples of mass fingerprint spectra of IFB-2 in *sma-5(n678)* showing the specific phosphate modification of phosphorylated peptides.

all instances (*Figure 4I*). Taken together, while the phosphorylation sites in the head domain of IFB-2 are not necessary for IF network morphogenesis per se, they may affect network properties such as subunit turnover that are not readily apparent from static pictures and may result in reduced network resilience leading to the decreased life span.

During the course of CRISPR mutagenesis, we isolated allele *ifb-2(kc27)*. This allele codes for an IFB-2 mutant containing not only the intended S2/S5/S7/S16−19>A mutations but a deletion of the region E31-A184, corresponding to the coil 1A, linker L1 and a large part of the coil 1B domain (see *Figure 5*). Interestingly, *ifb-2(kc27)* mutant animals showed an aberrant IFC-2 fluorescence pattern presenting a reduced though still apically localized IFC-2 distribution with additional, strongly fluorescent granules next to the CeAJ (*Figure 4F–F'*). The mutant protein could be detected by immunofluorescence in the intestine, albeit at highly reduced levels in the cytoplasm, at the CeAJ and in CeAJ-associated granules (*Figure 4—figure supplement 1A-B'*). The truncated IFB-2a and IFB-2c proteins of about 50 kDa were identified in immunoblots (*Figure 4—figure supplement 1C*). Significant effects on progeny and life expectancy were also noted (*Figure 4H–I*). We therefore propose that the aminoterminal region of IFB-2 up to A184 has an important function in network formation but that mutants lacking this region still retain the ability to enrich apically at the CeAJ.

Together, we conclude that altered phosphorylation status of the phosphorylation sites in the head domain are not solely responsible for the phenotype observed in SMA-5.

## Loss of SMA-5 induces hyperphosphorylation of multiple serine and tyrosine residues throughout the IFB-2 molecule

To find out whether phosphorylation sites other than those in the head domain may be regulated in *sma-5* mutants, we performed mass spectrometry on protein fractions obtained from synchronized wild-type N2 and *sma-5(n678)* adult animals (*Figure 5*, *Figure 5—figure supplement 1*). We identified eight different serine and tyrosine phosphorylation sites spread over the entire rod and tail domains. Remarkably, phosphopeptides containing S308, S422, Y253, Y335, and Y491 were only found in *sma-5(n678)*. Furthermore, phosphorylated S92 was detected in all four mutant samples and only in one

of four wild-type samples. An equal frequency of phosphopeptide detection was observed in mutant and wild-type samples for S55 and S383 (4/4 and 2/4 samples, respectively).

In summary, we can conclude that loss of SMA-5 is accompanied by hyperphosphorylation of IFB-2 affecting multiple phosphorylation sites. Given the large combinatorial realm of these phosphorylation sites and the likelihood that co-polymerizing intestinal IF polypeptides are also hyperphosphorylated, it will be very difficult or even impossible to unequivocally link hyperphosphorylation of defined sites in intestinal IFs to compromised *sma-5* activity. Nevertheless, our observations prove that SMA-5 is involved in regulating IFB-2 phosphorylation. They also indicate that IFB-2 is unlikely a direct target of SMA-5. Further experiments are needed to unravel how SMA-5 acts on IFB-2 phosphorylation, which may be accomplished by regulating the activity or recruitment of a phosphatase or by an even more indirect mechanism.

### *ifb-2(kc14)* partially rescues the *ifo-1(kc2)* and *bbln-1(mib70)* phenotypes

The above findings suggested that the rescue function of IFB-2 deletion in *sma-5(n678)* was caused by removal of the densely packed IFs. To test the hypothesis that perturbed IF network organization per se exerts adverse effects, we studied another paradigm, namely the *ifo-1(kc2)* phenotype, which is characterized by prominent IF polypeptide-containing junctional aggregates (*Carberry et al., 2012*). To this end, we crossed *ifo-1(kc2)* with *ifb-2(kc14)*. As predicted, the double mutant lacked the junctional IF aggregates altogether (*Figure 6A–D"*). Occasionally, a faint but largely reduced remnant endotube structure was detectable, often in the vicinity of the CeAJ. Furthermore, double mutants had an almost normal intestinal lumen and microvilli (*Figure 6D–D"*, *Figure 1—figure supplement 2* for quantification of luminal dilation). Additional analyses showed a significantly reduced time of development of *ifo-1(kc2);ifb-2(kc14)* in comparison to *ifo-1(kc2)*, which was, however, still significantly retarded in comparison to *ifb-2(kc14)* or wild-type N2 (*Figure 7A–E*). Moreover, analysis of body length revealed a significant rescue of the *ifo-1(kc2)* growth defect in the absence of the IFB-2 protein (*Figure 1—figure supplement 1*).

In a next set of experiments, we analyzed the rescuing efficiency of IFB-2 depletion in *bbln-1* mutants, which present a structural phenotype similar to but not identical to that encountered in *sma-5* mutants (*Remmelzwaal et al., 2021*). It had been shown recently that depletion of IFB-2 rescues the intestinal cytoplasmic invagination phenotype of *bbln-1* mutants (*Remmelzwaal et al., 2021*, *Figure 1—figure supplement 2* for quantification of luminal dilation). We now tested whether systemic dysfunctions also occur in *bbln-1* mutants and whether they can be rescued by IFB-2 depletion. We found that the *bbln-1(mib70)* loss-of-function mutants exhibited systemic dysfunctions as determined by body size and time of development (*Figure 1—figure supplement 1*, *Figure 8A*). Although BBLN-1 expression is, in contrast to IFO-1 and SMA-5, not limited to the intestine, the systemic dysfunctions were less pronounced than in *sma-5(n578)* or *ifo-1(kc2)*. The *ifb-2(kc14)* knockout allele rescued the reduced body size and prolonged time of development of *bbln-1(mib70)* (*Figure 1—figure supplement 1*, *Figure 8A*). The remaining growth defect and developmental retardation, however, were still slightly increased not only in comparison to the wild type but also to single *ifb-2(kc14)* mutants. To define the alterations in development more precisely, we analyzed the different developmental stages every day after hatching. The results shown in *Figure 8B–E* highlight the high degree of similarity in the developmental time course of the single *ifb-2(kc14)* and double *bbln-1(mib70);ifb-2(kc14)* mutants. Both develop more slowly than the wild type and much faster than *bbln-1(mib70)* animals.

*Figure 9* summarizes the morphological observations in the intestine for the three types of IF regulators together with the rescue phenotype resembling that of the *ifb-2(kc14)* mutant. It draws attention to the major conclusion that perturbed IF networks affect the structural and functional integrity of the intestinal IF network, the removal of which alleviates the mutant phenotypes.

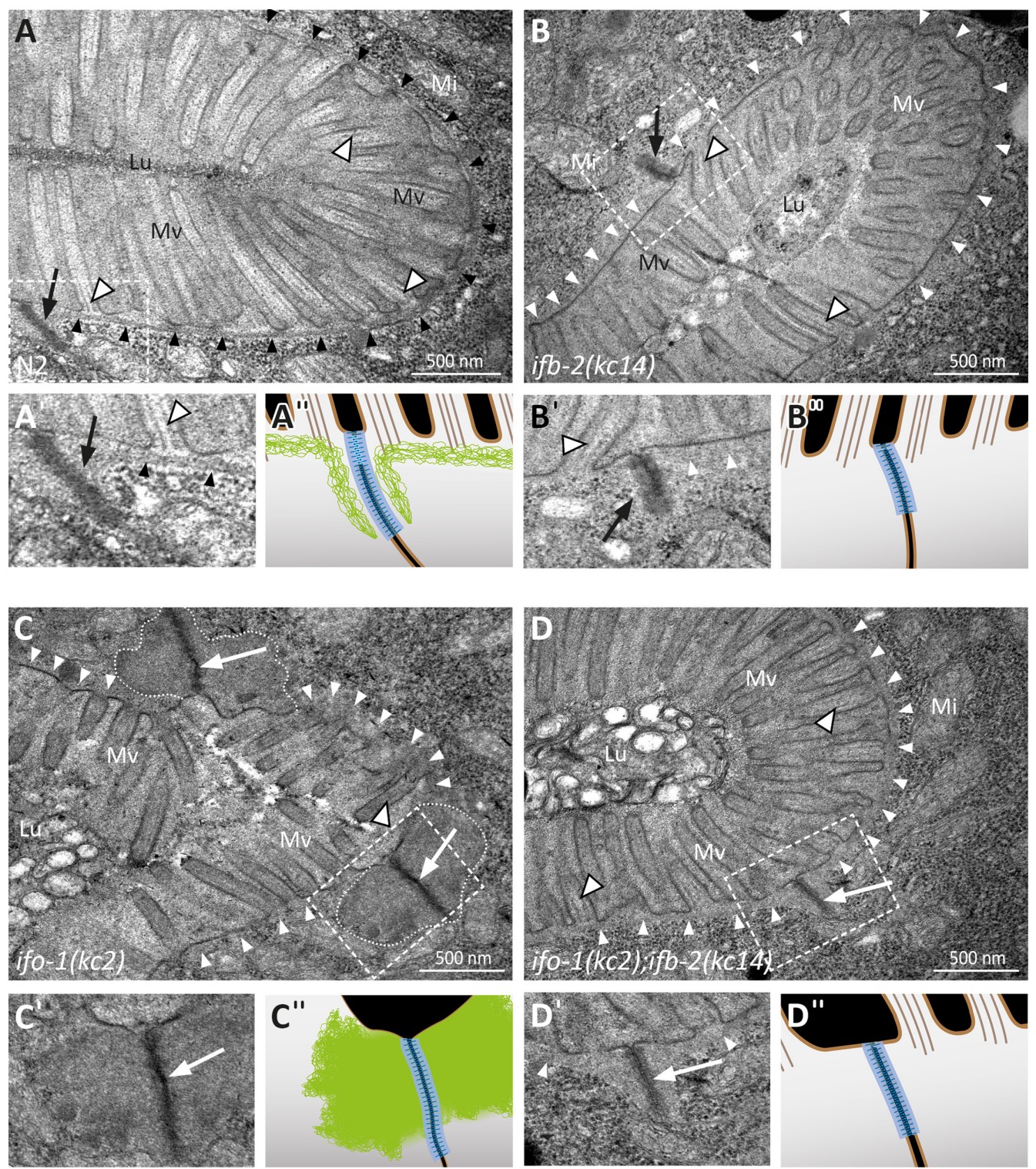

**Figure 6.** Depletion of IFB-2 partially rescues the *ifo-1(kc2)* phenotype. (**A–D**) The electron micrographs show a comparison of the intestinal cell apices surrounding the lumen (Lu) of wild-type N2 (**A**, enlarged section in **A'**, corresponding scheme in **A''**), *ifb-2(kc14)* (**B**, enlarged section in **B'**, corresponding scheme in **B''**), *ifo-1(kc2)* (**C**, enlarged section in **C'**, corresponding scheme in **C''**), and *ifo-1(kc2);ifb-2(kc14)* (**D**, enlarged section in **D'**, corresponding scheme in **D''**). Note the distinct endotube in N2 (black arrowheads) and its absence in *ifb-2(kc14)*, *ifo-1(kc2)*, and *ifo-1(kc2);ifb-2(kc14)* (white arrowheads). The pathognomonic large junctional aggregates of *ifo-1(kc2)* are delineated by broken white lines. Note also the improved brush border morphology in (**D**) compared to (**C**) (Mv, microvilli). Arrows, *C. elegans* apical junction (CeAJ); outlined white arrowheads, microvillar actin bundles; Mi, mitochondrion.

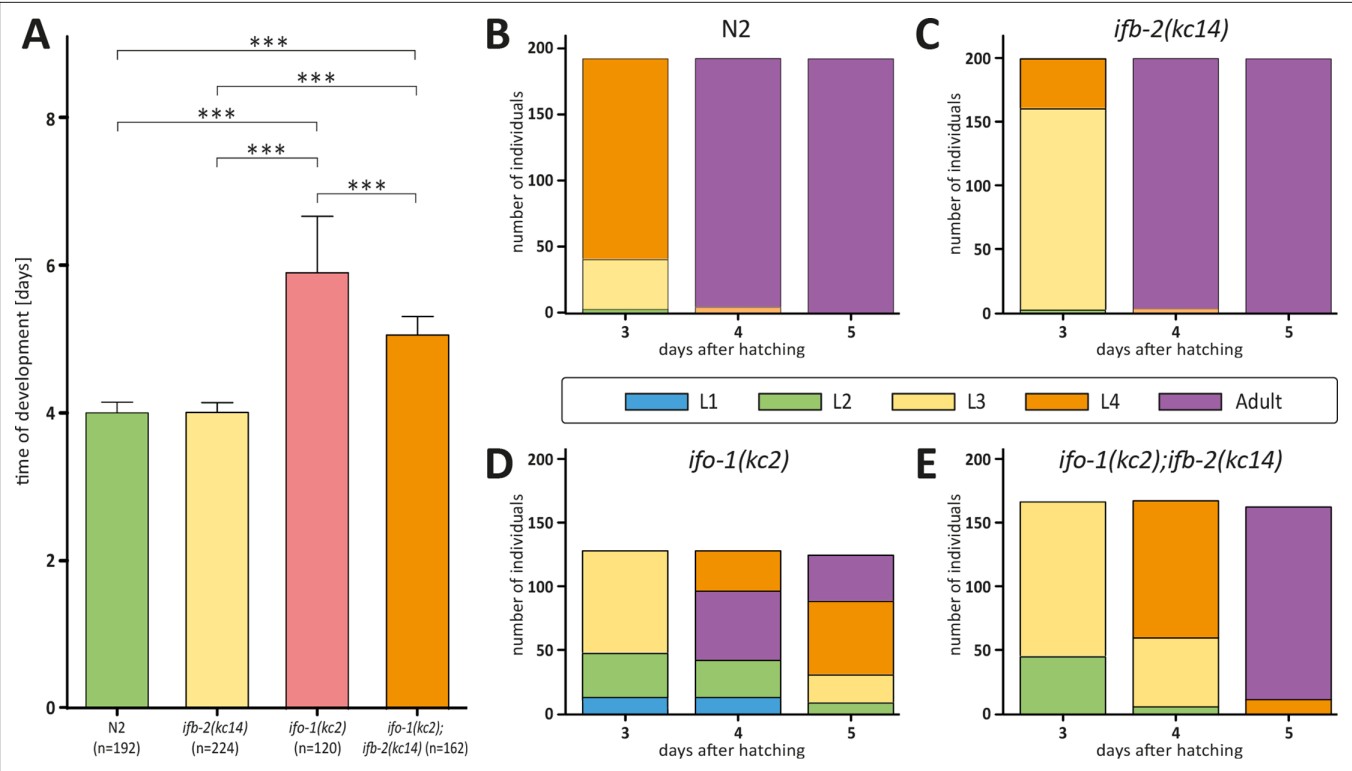

**Figure 7.** Depletion of IFB-2 partially rescues the *ifo-1(kc2)* growth defects. (**A–E**) The histograms depict the time of development in (**A**) and the number of staged worms at different times after hatching in N2 (**B**), *ifb-2(kc14)* (**C**), *ifo-1(kc2)* (**D**), and *ifo-1(kc2);ifb-2(kc14)* (**E**). Note the partial rescue in the double mutants. (N2: 4.0±0.1 days; *ifb-2(kc14)*: 4.0±0.1 days; *ifo-1(kc2)*: 5.9±0.8 days; *ifo-1(kc2);ifb-2(kc14)*: 5.1±0.3 days; ***p<0.0001).

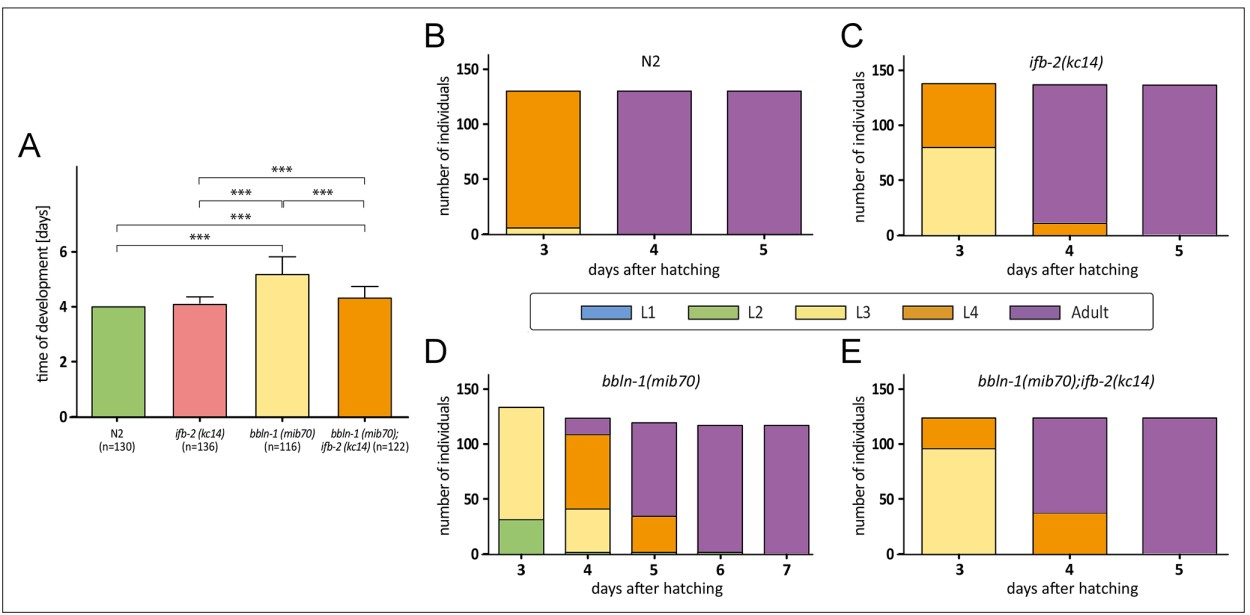

**Figure 8.** Depletion of IFB-2 partially rescues the *bbln-1(mib70)* phenotype. (**A–E**) The histograms depict the time of development in (**A**) and the number of staged worms at different times after hatching in N2 (**B**), *ifb-2(kc14)* (**C**), *bbln-1(mib70)* (**D**), and *bbln-1(mib70);ifb-2(kc14)* (**E**). Note the partial rescue in the double mutants (N2: 4.0 days; *ifb-2(kc14)*: 4.1±0.3 days; *bbln-1(mib70)*: 5.2±0.6 days; *bbln-1(mib70);ifb-2(kc14)*: 4.3±0.5 days; ***p<0.0001).

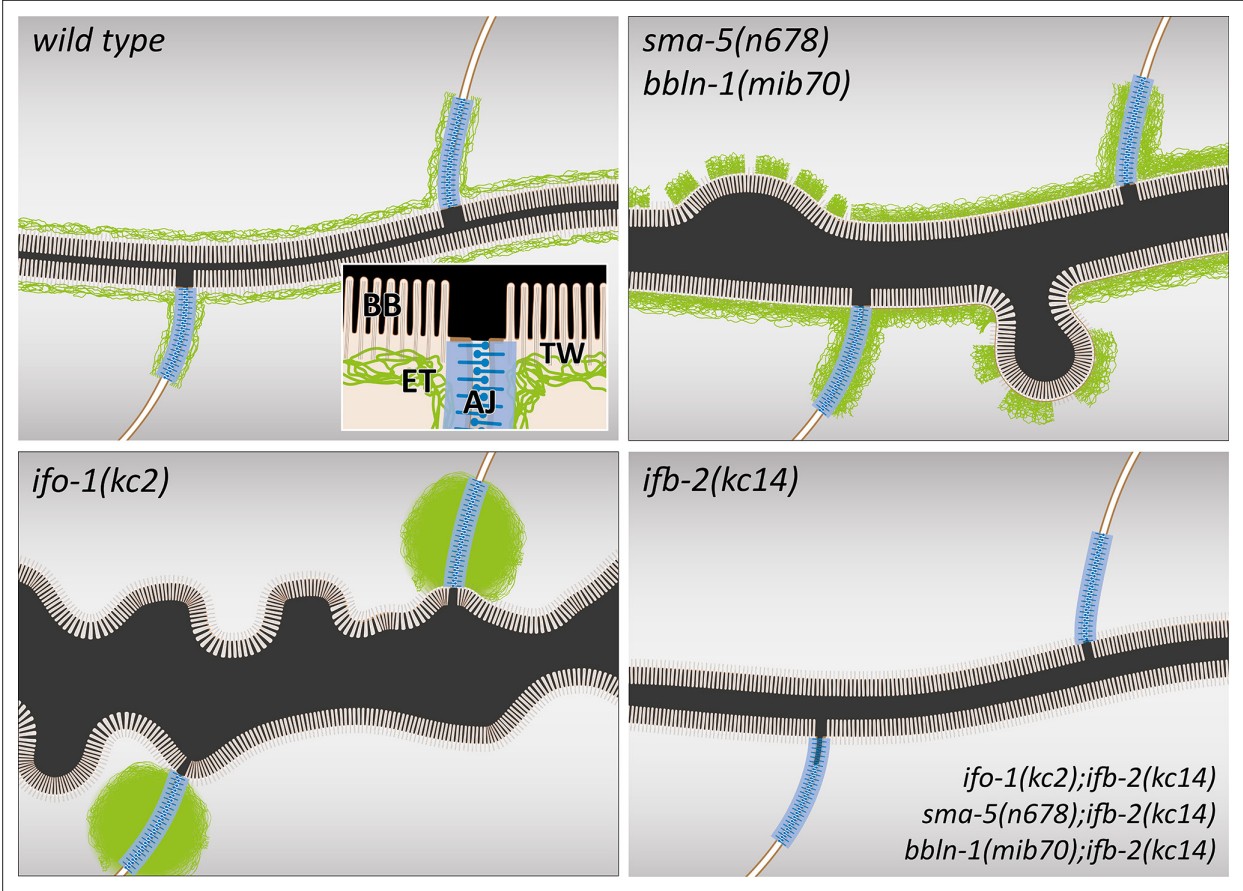

**Figure 9.** The schematic drawings highlight changes in the intestinal lumen and intermediate filament (IF) network organization in the different mutant backgrounds. The wild-type scheme (upper left) depicts the adluminal brush border (BB) consisting of microvilli with bundled actin filaments, the terminal web (TW) with traversing microvillar actin rootlets that rest on the IF-rich endotube (ET) together with the *C. elegans* apical junction (AJ), which serves as an anchorage site for the IF network. Note the luminal enlargement and cytoplasmic invaginations in *sma-5(n678)* and *bbln-1(mib70)* presenting a thickened and discontinuous endotube with slightly disordered microvilli (upper right). *ifo-1(kc2)* is characterized by endotube loss and formation of intermediate filament aggregates at the *C. elegans* apical junction (lower left). Luminal widening, cytoplasmic invaginations and microvillar disorder are also observed. The mildest phenotype is detectable in *ifb-2(kc14)* and in the double mutants *ifo-1(kc2);ifb-2(kc14)*, *sma-5(n678);ifb-2(kc14)*, and *bbln-1(mib70);ifb-2(kc14)*.

## Discussion

Exploiting rapid *C. elegans* genetic screening, we observed that the complex phenotypes induced by mutation of the MAPK orthologue SMA-5 can be rescued by deletion of the cytoskeletal IF protein IFB-2. This included rescue of structural defects (cytoplasmic invagination and lumen dilatation), developmental and growth defects, as well as oxidative stress resilience. The obtained rescue levels coincided precisely with the mild phenotype of single *ifb-2(kc14)* mutants. Remarkably, osmotic stress resilience could not be rescued suggesting that this loss-of-function is tightly coupled to the absence of the endotube. In accordance, it has been suggested that osmotic stress resilience is a fundamental function of cytoplasmic IFs (*D'Alessandro et al., 2002*, *Pekny and Lane, 2007*). Osmotic challenges are of particular relevance to the intestine and its exposure to microbial toxins. Our findings furthermore provide strong in vivo evidence that the *sma-5* mutant phenotype is caused by the presence of IFB-2-containing pathological assemblies. This conclusion was confirmed in *ifo-1* and *bbln-1* mutants. Removal of the pathological IFB-2 assemblies also rescued complex biological functions. We therefore conclude that the deranged intestinal IF cytoskeletons are associated with a gain-of-toxic function exerting negative effects, which have detrimental consequences for cell and tissue function and thereby adversely affect growth and reproduction of the entire organism. The scenario is reminiscent

of that described in vertebrate disease paradigms, in which IF depletion has positive effects on disease outcome (review in *Ridge et al., 2022*).

The fact that removal of the very differently deranged IF networks, that is the thick subapical slabs in *sma-5(n678)* and *bbln-1(mib70)* and the large junctional aggregates in *ifo-1(kc2)*, rescued the phenotypes in all instances is quite remarkable since the mechanical and structural dysfunctions appear to be fundamentally different in the different mutant backgrounds. They manifest in *sma-5* and *bbln-1* mutants as prominent cytoplasmic invaginations and primarily as luminal widening in *ifo-1* mutants (see also *Geisler et al., 2016*; *Carberry et al., 2012*; *Remmelzwaal et al., 2021*). Furthermore, the absence of an endotube in *ifo-1* mutants predicts that the force equilibrium is affected differently from that in *sma-5* and *bbln-1* mutants with the locally thickened endotube. This indicates that restored mechanics alone are not sufficient to explain the phenotypic restoration in the double mutants. Thus, other non-mechanical functions must be attributable to the abnormal IF assemblies. It is interesting in this context that IFs may serve as signaling platforms by providing a large scaffold capable of sequestering and positioning signaling molecules that can be recruited by weak interactions and can be released, for example, by protein modification or structural changes of the IF cytoskeleton (review in *Bott and Winckler, 2020*; *Coulombe and Omary, 2002*; *Magin et al., 2007*). Future experiments will be needed to show whether such a scaffolding function is compromised in *sma-5*, *bbln-1,* and *ifo-1* mutants by either sequestering or setting free regulatory factors that modulate pathways needed for normal growth, development, and stress responses.

By focusing on IF phosphorylation, we could show that IFs are hyperphosphorylated in *sma-5(n678)* animals and not hypophosphorylated as would be expected in case of a direct SMA-5/IF interaction. This implies that SMA-5 regulates IF phosphorylation indirectly. It remains to be shown whether this is mediated by controlling the activity or the recruitment of a phosphatase. This scenario predicts that activation of this yet unidentified phosphatase would restore normal IF phosphorylation levels and IF network morphogenesis in *sma-5(n678)* mutants. An alternative explanation would invoke an even more indirect mechanism whereby SMA-5 phosphorylates a factor that affects IF assembly, which in turn modulates IFB-2 phosphorylation. We do not know whether IF phosphorylation is also altered in *bbln-1(mib70)* and *ifo-1(kc2)* and whether IF hyperphosphorylation is per se able to induce IF network aggregation. The notion of phosphorylation-dependent IF network morphogenesis, however, is in line with numerous publications that have implicated phosphorylation in the regulation of IF assembly and disassembly (reviews in *Sawant and Leube, 2017*; *Nishimura et al., 2019*). It is furthermore supported by the observation that IF phosphoepitopes are increased in IF aggregates (*Sawant and Leube, 2017*; *Sawant et al., 2018*; *Wöll et al., 2007*; *Battaglia et al., 2019*; *Binukumar et al., 2013*). Despite the identification of dominant phosphoregulatory sites it has been very difficult to define single sites that are exclusively responsible for a given structural or functional phenotype. This is in agreement with the current observations identifying multiple sites that are spread throughout the IFB-2 molecule. The future challenge is therefore to understand, in quantitative terms, how the multiple phosphorylation sites act together in modulating IF network formation and function. While this is technically difficult to achieve, it endows the cell with endless options to fine-tune its cytoskeleton in a highly context-dependent fashion and to connect the fine-tuning to multiple signaling pathways. We posit that the intestine-specific MAPK SMA-5 is linked to one or more of these signaling pathways, which affect the balance between phosphorylation and dephosphorylation of intestinal IFs and possibly other mediators. Similarly, IFO-1 may be linked to a similar pathway since it was initially identified as a toxin-regulated target of MAPK (*ttm-4*; *Kao et al., 2011*).

Obviously, further work is needed to fully understand both the relevant targets of SMA-5 and the role of phosphorylation in IF aggregate formation in order to elucidate the exact mechanism of aggregate suppression. However, our findings are likely relevant for the multiple aggregate-forming human diseases that involve IF polypeptides (e.g., *Coulombe et al., 2009*; *Chamcheu et al., 2011*; *Yoshida and Nakagawa, 2012*; *Clemen et al., 2013*; *Gentil et al., 2015*; *Didonna and Opal, 2019*). They mandate a careful evaluation of possible toxic functions of the perturbed IF networks, which may not be limited to the local cell and tissue environment but may have systemic consequences.

# Materials and methods

## Key resources table

| Reagent type (species) or resource | Designation | Source or reference | Identifiers | Additional information |
|---|---|---|---|---|
| Antibody | Anti-IFB-2 (mouse monoclonal) | Developmental Studies Hybridoma Bank | AB_528311 MH33 | (1:100-1:1000), (*Francis and Waterston, 1991*) |
| Antibody | Anti-mouse IgG coupled to horseradish peroxidase (goat polyclonal) | DAKO | #P0447 | (1:5000) |
| Antibody | Anti-GFP (rabbit polyclonal) | Invitrogen | #A-11122 | (1:1000) |
| Antibody | Anti-mouse IgG coupled to Alexa Fluor 488 (goat polyclonal) | Invitrogen | #A-11029 | (1:200) |
| Antibody | Anti-mouse IgG coupled to Alexa Fluor 555 (goat polyclonal) | Invitrogen | #A-21424 | (1:200) |
| Strain, strain background (*Escherichia coli*) | OP50 | CGC | N/A | Strain can be obtained from the Caenorhabditis Genetics Center |
| Strain, strain background (*Escherichia coli*) | HT115 | CGC | N/A | Strain can be obtained from the Caenorhabditis Genetics Center |
| Strain, strain background (*Escherichia coli*) | Vidal full-length HT115 RNAi feeding library | SourceBio-Science | 3320_Cel_ORF_RNAi | |
| Strain, strain background (*Escherichia coli*) | Ahringer fragment HT115 RNAi feeding library | SourceBio-Science | 3318_Cel_RNAi_complete | |
| Chemical compound, drug | Dextran, Texas Red, 70,000 MW | Invitrogen | #D1830 | |
| Chemical compound, drug | Alt-R S.p. Cas9 Nuclease V3 | IDT | #1081058 | |
| Strain, strain background (*Caenorhabditis elegans*) | Wild type (Bristol) | CGC | N2 | Strain can be obtained from the Caenorhabditis Genetics Center |
| Strain, strain background (*Caenorhabditis elegans*) | sma-5(n678)X | CGC | FK312 | Strain can be obtained from the Caenorhabditis Genetics Center |
| Strain, strain background (*Caenorhabditis elegans*) | kcIs6[ifb-2p::ifb-2a::cfp]IV | *Hüsken et al., 2008* | BJ49 | Strain can be obtained from the Leube lab |
| Strain, strain background (*Caenorhabditis elegans*) | ifo-1(kc2)IV | *Carberry et al., 2012* | BJ142 | Strain can be obtained from the Leube lab |
| Strain, strain background (*Caenorhabditis elegans*) | sma-5(n678)X;kcIs6[ifb-2p::ifb-2a::cfp]IV | *Geisler et al., 2016* | OLB18 | Strain can be obtained from the Leube lab |
| Strain, strain background (*Caenorhabditis elegans*) | ifb-2(kc14)II | *Geisler et al., 2019* | BJ309 | Strain can be obtained from the Leube lab |
| Strain, strain background (*Caenorhabditis elegans*) | erm-1(mib40[erm-1::AID::mCherry])I; Pelt-2::TIR::tagBFP-2::tbb-2–3'UTR (mib58[Pelt-2::TIR-1::tagBFP-2 -Lox511::tbb-2–3'UTR])IV;C15C7.5 (mib70[pC15C7.5::GFP1-3, C15C7.5 0-914del])X | *Remmelzwaal et al., 2021* | BOX415 | Strain can be obtained from the Boxem lab |
| Strain, strain background (*Caenorhabditis elegans*) | kcEx78[WRM0639B_D12(pRedFlp-Hgr) (ifc-1[24863] ::S0001_pR6K_Amp_2xTY1ce_ EGFP_FRT_ rpsl_neo_FRT_3xFlag) dFRT ::unc-119-Nat];unc-119(ed3)III | *Geisler et al., 2020* | BJ324 | Strain can be obtained from the Leube lab |
| Strain, strain background (*Caenorhabditis elegans*) | ifc-2(kc16)X | *Geisler et al., 2020* | BJ316 | Strain can be obtained from the Leube lab |

*Continued on next page*

*Continued*

| Reagent type (species) or resource | Designation | Source or reference | Identifiers | Additional information |
|---|---|---|---|---|
| Strain, strain background (*Caenorhabditis elegans*) | kcIs40[ifp-1p::ifp-1::egfp]IV | *Geisler et al., 2020* | BJ312 | Strain can be obtained from the Leube lab |
| Strain, strain background (*Caenorhabditis elegans*) | erm-1(mib40[erm-1::AID::mCherry]) I; ifd-2(mib94[GFP::ifd-2])X | *Remmelzwaal et al., 2021* | BOX614 | Strain can be obtained from the Boxem lab |
| Strain, strain background (*Caenorhabditis elegans*) | sma-5(n678)X;ifb-2(kc14)II | This study | BJ346 | Strain can be obtained from the Leube lab |
| Strain, strain background (*Caenorhabditis elegans*) | ifo-1(kc2)IV;ifb-2(kc14)II | This study | BJ328 | Strain can be obtained from the Leube lab |
| Strain, strain background (*Caenorhabditis elegans*) | bbln-1(mib70) | This study | BJ411 | Strain can be obtained from the Leube lab |
| Strain, strain background (*Caenorhabditis elegans*) | bbln-1(mib70);ifb-2(kc14) | This study | BJ412 | Strain can be obtained from the Leube lab |
| Strain, strain background (*Caenorhabditis elegans*) | erm-1(mib40[erm-1::AID::mCherry])I; Pelt-2::TIR::tagBFP-2 ::tbb-2–3'UTR (mib58[Pelt-2:: TIR-1:: tagBFP-2-Lox511::tbb-2–3'UTR])IV; C15C7.5(mib70[pC15C7.5 ::GFP1-3, C15C7.5 0-914del])X;ifb-2(kc14)II | This study | BJ364 | Strain can be obtained from the Leube lab |
| Strain, strain background (*Caenorhabditis elegans*) | ifd-2(mib94[GFP::ifd-2])X | This study | BJ405 | Strain can be obtained from the Leube lab |
| Strain, strain background (*Caenorhabditis elegans*) | ifb-2(mib155[S2D;S5D; S7D;S16-19D])II; ifc-2a/e::yfp(kc16)X | This study | BOX793 | Strain can be obtained from the Leube lab |
| Strain, strain background (*Caenorhabditis elegans*) | ifb-2(kc22[S2D;S5D; S7D;S16-19D;S24D; S35D])II;ifc-2a/e::yfp(kc16)X | This study | BJ427 | Strain can be obtained from the Leube lab |
| Strain, strain background (*Caenorhabditis elegans*) | ifb-2(mib156[S2A;S5A; S7A;S16-19A])II; ifc-2a/e::yfp(kc16)X | This study | BOX794 | Strain can be obtained from the Leube lab |
| Strain, strain background (*Caenorhabditis elegans*) | ifb-2(kc26[S2A;S5A; S7A; S16-19A;S24A;S35A])II; ifc-2a/e::yfp(kc16)X | This study | BJ431 | Strain can be obtained from the Leube lab |
| Strain, strain background (*Caenorhabditis elegans*) | ifb-2(kc27[S2A;S5A;S7A; S16-19A;E31-A184 deleted])II; ifc-2a/e::yfp(kc16)X | This study | BJ432 | Strain can be obtained from the Leube lab |
| Strain, strain background (*Caenorhabditis elegans*) | ifb-2(mib169[S2-M43 deleted])II;ifc-2a/e::yfp(kc16)X | This study | BOX827 | Strain can be obtained from the Leube lab |
| Strain, strain background (*Caenorhabditis elegans*) | ifd-1(mib95[mCherry::ifd-1])X | This study | BOX540 | Strain can be obtained from the Boxem lab |
| Sequence-based reagent | See *Supplementary file 1* and "RNAi and body length determination" section for sequence details | IDT | | |
| Recombinant DNA reagent | L4440 | Addgene | #1654 | |
| Recombinant DNA reagent | ifd-1 (L4440) | *Remmelzwaal et al., 2021* | pSMR35 | Reagent can be obtained from the Boxem lab |
| Recombinant DNA reagent | ifp-1 (L4440) | *Remmelzwaal et al., 2021* | pSMR34 | Reagent can be obtained from the Boxem lab |
| Recombinant DNA reagent | rol-6(su1006) | *Mello et al., 1991* | pRF4 | Reagent can be obtained from the Leube lab |
| Software, algorithm | ImageJ | Rasband, W.S. (NIH) | RRID: SCR_003070 | |

*Continued on next page*

*Continued*

| Reagent type (species) or resource | Designation | Source or reference | Identifiers | Additional information |
|---|---|---|---|---|
| Software, algorithm | Zen Blue | Zeiss | RRID:SCR_013672 | |
| Software, algorithm | GraphPad Prism | GraphPad | RRID: SCR_002798 | |
| Software, algorithm | SnapGene | Insightful Science | RRID: SCR_015052 | |
| Software, algorithm | Clone Manager | Scientific & Educational Software | RRID:SCR_014521 | |
| Software, algorithm | Bruker Bio-Tool 3.2 and the Mascot 2.3 search engine | Matrix Science Ltd | N/A | |

## *C. elegans* strains and bacteria

Wild-type strain N2, strain FK312 *sma-5(n678)X,* and OP50 bacteria were obtained from the Caenorhabditis Genetics Center (CGC; University of Minnesota, MN, USA). Strains BJ49 *kcIs6[ifb-2p::ifb-2a::cfp]IV* (**Hüsken et al., 2008**), BJ142 *ifo-1(kc2)IV* (**Carberry et al., 2012**), OLB18 *sma-5(n678)X;kcIs6[ifb-2p::ifb-2a::cfp]IV* (**Geisler et al., 2016**), BJ309 *ifb-2(kc14)II* (**Geisler et al., 2019**), BOX415 *erm-1(mib40[erm-1::AID::mCherry])I;Pelt-2::TIR::tagBFP-2::tbb-2–3'UTR (mib58[Pelt-2::TIR-1::tagBFP-2-Lox511::tbb-2–3'UTR])IV;C15C7.5(mib70[pC15C7.5::GFP1-3, C15C7.5 0-914del])X* (**Remmelzwaal et al., 2021**), BJ324 *kcEx78[WRM0639B_D12(pRedFlp-Hgr)(ifc-1[24863]::S0001_pR6K_Amp_2x-TY1ce_EGFP_FRT_rpsl_neo_FRT_3xFlag)dFRT::unc-119-Nat];unc-119(ed3)III* (**Geisler et al., 2020**), BJ316 *ifc-2(kc16)X* (**Geisler et al., 2020**), BJ312 *kcIs40[ifp-1p::ifp-1::egfp]IV* (**Geisler et al., 2020**) and BOX614 *erm-1(mib40[erm-1::AID::mCherry]) I; ifd-2(mib94[GFP::ifd-2])X* **Remmelzwaal et al., 2021** have been described. Strain FK312 was crossed with strain BJ309 to obtain strain BJ346 *sma-5(n678)X;ifb-2(kc14)II*. Strain BJ328 *ifo-1(kc2)IV;ifb-2(kc14)II* was generated by crossing strain BJ142 with strain BJ309. Strains BJ411 *bbln-1(mib70)* and BJ412 *bbln-1(mib70);ifb-2(kc14)* were generated by crossing BOX415 with BJ309 resulting in strain BJ364 *erm-1(mib40[erm-1::AID::mCherry])I;Pelt-2::TIR::tagBFP-2::tbb-2–3'UTR (mib58[Pelt-2::TIR-1::tagBFP-2-Lox511::tbb-2–3'UTR])IV;C15C7.5(mib-70[pC15C7.5::GFP1-3, C15C7.5 0-914del])X;ifb-2(kc14)II*, which was subsequently crossed with N2. Strain BJ405 *ifd-2(mib94[GFP::ifd-2])X* was generated by crossing BOX614 with N2.

## Suppressor screen

Ten cm diameter agar plates containing normal nematode growth medium (NGM) were prepared by autoclaving a solution containing 3 g NaCl (Carl Roth, Karlsruhe, Germany, #3957.1), 2.5 g Bacto peptone (BD BioSciences, Heidelberg, Germany, #211820) and 18.75 g Bacto agar (BD BioSciences, #90000-764) per 1 l and pouring the solution after addition of 1 ml cholesterol (5 mg/ml in ethanol), 0.5 ml 1 M CaCl$_2$, 1 ml 1 M MgSO$_4$, and 25 ml 1 M KH$_2$PO$_4$ (pH 6.0). After solidification, OP50 were placed on top of the agar and plates were incubated overnight at 37°C. They were then used for growing OLB18. Adult animals were bleached with a solution containing 170 µl 12% NaOCl, 100 µl 4 M NaOH per 1 ml PBS (Biochrom, Berlin, Germany) and the resulting embryos were grown on new plates. The resulting synchronized L4 larvae were washed off with PBS and were then centrifuged in 15 ml Falcon tubes at 340×*g* for 2 min. Five ml of the pelleted worms were mixed with 50 µl of 50 mM *N*-ethyl-*N*-nitrosourea (Sigma-Aldrich, Munich, Germany, N3385-1G) and the suspension was incubated for 4 hr at room temperature on a rotating shaker. Two washing steps with PBS followed before the mutagenized worms were resuspended in PBS. One hundred L4 animals were placed per agar plate containing enriched NGM supplemented with 5 g/l Difco yeast extract (Thermo Fisher Scientific, MA, USA, #210929) (total of 200 plates). The plates were then incubated at 20°C. It was ensured that the F2 generation was completely laid on these plates before food was used up. Then 1.5×1.5 cm$^2$ pieces were cut out and placed on new plates, which were again incubated at 20°C until food was completely used up. After repeating these steps three times, the development of the mutant lines compared to OLB18 was monitored using a stereomicroscope. Criteria for suppressor activity were body size and developmental stage. If both were rescued, lines were subjected to further selection rounds. Only stable lines were subsequently examined by light microscopy for rescue of invaginations of the apical intestinal membrane. The mutant line, which met all three criteria best, was named strain BJ334 (*sma-5(n678)X;ifb-2(kc20)II*) and was backcrossed twice with OLB18 resulting in strain BJ355 (*sma-5(n678)X;ifb-2(kc20)II*). To determine the mutation of allele *kc20 ifb-2* DNA was amplified in three parts using three primer pairs outside the protein coding regions (oSMR66 (ggtgttggtttttttaactgc

tg) and oSMR68 (acacacccatttcctccaga), oSMR67 (tctggaggaaatgggtgtgt), and oSMR70 (tccttgcggata cactctga), oSMR69 (tcggtagctataaccgcttca), and oSMR71 (caaggaaaggattcaatgggc)). The DNA was purified and analyzed by Sanger sequencing using the same primers as for amplification.

## CRISPR-Cas9

All alleles were made using homology-directed repair of CRISPR-Cas9-induced DNA double strand breaks, using microinjected Cas9 ribonucleoprotein complexes and linear ssDNA oligo repair templates, similar to the approach described by *Ghanta et al., 2021*. We used 250–700 ng/µl Cas9, a Cas9/crRNA ratio of 3.0–4.5, and the pRF4 (*rol-6(su1006)*) co-injection marker (*Mello et al., 1991*). Repair templates included, when appropriate, silent mutations to prevent recutting of repaired loci by Cas9. To verify the edits, the insertion sites were PCR amplified and sequenced by Sanger sequencing. A summary of DNA/RNA-based reagents is shown in *Supplementary file 1*.

## Phosphoproteomics

One hundred and twenty young adults were picked for each sample of strains N2 and FK312. They were placed in 30 µl dH$_2$O each and subsequently washed two times followed by freezing in liquid nitrogen and storage at –80°C until further use. To disrupt the cuticle, the samples were subjected to five cycles of rapid thawing on a heating block at 100°C and freezing in liquid nitrogen. Samples were subsequently homogenized by up and down pipetting five times through a 30 G hypodermal syringe (BD Medical, Heidelberg, Germany, #MPC B3324826). After addition of 7.5 µl 5× Laemmli loading buffer (15 ml stacking gel buffer [0.15 M Tris-Cl, 0.1% SDS, pH: 6,8], 12.5 ml glycerine, 2.5 ml β-mercaptoethanol, 2.5 g SDS, some bromophenol blue), the samples were incubated for 5 min at 100°C. Polypeptides were separated by electrophoresis in an 10% sodium dodecyl sulfate (SDS) poly-acrylamide gel followed up by Coomassie Brilliant Blue staining. The region with proteins between 50 kDa and 150 kDa was manually excised and further processed for matrix-assisted-laser-desorption/ionization-time-of-flight-mass spectrometry (MALDI-TOF-MS) as previously described (*Rueth et al., 2015*; *Kork et al., 2018*). Briefly, the protein plugs were incubated with ammonium bicarbonate (50 mmol/l) and 0.05% w/c trypsin for 24 hr at 37°C. The resulting tryptic peptides were desalted and concentrated using ZipTip$_{c18}$ technology (Millipore, Billerica, MA, USA) and eluted with 80% acetonitrile directly onto the (MALDI) target plate (MTP-AnchorChip 400/384; Bruker-Daltonic) using alpha-cyano-4-hydroxycinnamic acid as matrix. The subsequent MS analyses were performed using a MALDI-time of flight/time of flight (TOF/TOF) mass spectrometer (Ultraflex III and Rapiflex; Bruker-Daltonic, Bremen, Germany). MS/MS fragments were analyzed using Lift-option of the mass spec-trometer. Calibrated and annotated spectra were subjected to the database search Swiss-Prot (http://www.expasy.org/) utilizing the software tool 'Bruker Bio-Tool 3.2' and the 'Mascot 2.3 search engine' (Matrix Science Ltd, London, UK). A total of four samples per strain were analyzed.

## Microscopy

Light microscopy was performed with a Zeiss (Jena, Germany) apotome in combination with a ZeissAxioCamMRm camera.

For electron microscopy worms were either chemically fixed (*Figure 5*) or cryofixed at high pres-sure (*Figure 1*).

### Chemical fixation

Young adult animals (40 for each strain) were submerged in freshly prepared fixation solution containing 2.5% glutaraldehyde (using 25% (wt/vol) glutaraldehyde from Carl Roth), 1% (wt/vol) paraformalde-hyde (using a stock solution with 0.4 g paraformaldehyde dissolved in 10 ml 0.1 M sucrose with 0.6 µl 10 N sodium hydroxide [all from Sigma-Aldrich]), 0.05 M cacodylate buffer (using dimethylarsinic acid sodium salt trihydrate from Merck and HCl for pH adjustment to 6.4–7.4) at room temperature in a glass staining block. Each worm was cut through at the anterior and posterior end with a scalpel. The fixation solution was exchanged two times with 1.5–2 hr incubation at room temperature in between. After the third replacement of the fixation solution an overnight incubation followed at 4°C in a moist chamber. The samples were then transferred to 0.2 M cacodylate buffer and incubated for 3×10 min in this buffer with buffer changes in between. Samples were then incubated for 4 hr in 0.1 M caco-dylate buffer containing 1% (wt/vol) OsO$_4$ (Paesel-Lorei, Frankfurt/Main, Germany) and 0.5% (wt/vol)

K$_3$[Fe(CN)$_6$] (Carl Roth), followed by 3×10 min in 0.1 M cacodylate buffer. Sometimes samples were incubated overnight at 4°C in a moist chamber in 0.1 M cacodylate buffer. The following incubations ensued at room temperature: 3×10 min in 0.1 M maleic acid buffer (Sigma-Aldrich; pH 6), 2 hr in the dark in 0.5% (wt/vol) uranyl acetate (EMS, Hatfield, PA, USA) dissolved in 0.05 N maleic acid buffer (pH 5.2), 3×5 min in 0.1 M maleic acid buffer, 3×5 min in double distilled water, 5 min in 20% ethanol, 5 min in 30% ethanol, 3×10 min in 50% ethanol, 2×15 min in 75% ethanol, 2×15 min in 95% ethanol, 3×10 min in 100% ethanol, 10 min in a 1:1 ethanol/acetone mixture, and 2×10 min in acetone. Finally, samples were embedded in araldite (Agar Scientific, Stansted, UK). To this end samples were first placed overnight at 4°C in a 3:1 mixture of acetone/araldite with 1.5% (vol/vol) DMP30 (Agar Scientific) followed by incubation steps at room temperature: 1 hr in a 1:1 mixture of acetone/araldite with 1.5% DMP30, 2 hr in a 1:3 mixture of acetone/araldite with 1.5% DMP30, and 2 days in araldite with 2% DMP30. The final polymerization was done at 60°C for 2 days in silicone molds with pre-polymerized araldite at the bottom. Seventy-five nm sections were prepared using a Leica (Wetzlar, Germany) Reichert Ultracut S microtome. They were contrasted for 4 min in uranyl acetate and 3 min in lead citrate and finally imaged at 60 kV in a Zeiss EM 10.

## High-pressure freezing

Young adult animals were transferred into a 100 µm deep membrane carrier containing 20% bovine serum albumin in M9 worm buffer (22 mM KH$_2$PO$_4$, 42 mM Na$_2$HPO$_4$, 86 mM NaCl, 1 mM MgSO$_4$) and then high-pressure frozen in a Leica EM Pact high-pressure freeze. A minimum of five samples with 10–20 animals were frozen per experiment. Quick freeze substitution using 1% OsO$_4$, 0.2% uranyl acetate in acetone followed by epoxy resin embedding was performed as previously described (*McDonald and Webb, 2011*). Subsequently, 50 nm thick sections of the embedded samples were prepared using a Leica UC6/FC6 ultramicrotome. These were contrasted for 10 min in 1% uranyl acetate in ethanol and Reynolds lead citrate and recorded at 100 kV on a Hitachi H-7600 transmission electron microscope (Tokyo, Japan).

## Dextran feeding

Standard OP50 plates were inoculated with 0.3% Dextran Texas Red 70 kDa (wt/vol, Invitrogen, Waltham, MA, USA, #D1830). After incubation at room temperature overnight, plates were ready to use. Approximately 30–40 L4 larvae were transferred on each plate and incubated at 20°C for 24 hr. Animals were mounted on agar slides using 10% levamisole as anesthetic. Non-vital animals were censored. To obtain comparable data, the anterior half of the intestine was imaged in each case. Two independent experiments were performed. Number of n states the number of biological replicates. Experiments were not randomized and investigators were not blinded during experiments and result analysis.

## Immunoblotting

Sixty young adults of strains N2 and BJ432 were picked in 30 µl dH$_2$O each and frozen at –80°C. To disrupt the cuticle, the samples were then rapidly thawed on a heating block and three times sucked up and down through a 30 G hypodermal syringe (BD Medical, Heidelberg, Germany, #MPC B3324826). After addition of 7.5 µl 5× Laemmli loading buffer (15 ml stacking gel buffer [0.15 M Tris-Cl, 0.1% SDS, pH: 6,8], 12.5 ml glycerine, 2.5 ml β-mercaptoethanol, 2.5 g SDS, some bromophenol blue), the samples were incubated for 5 min at 100°C. Polypeptides were separated by electrophoresis in an 8% SDS polyacrylamide gel. Separated proteins were then transferred by wet tank blotting (100 V for 60 min) onto a polyvinylidene difluoride membrane (Merck, Darmstadt, Germany, #IPVH00010). The membrane was blocked with Roti-Block (2 hr at room temperature; Carl Roth, #A151.2) and incubated overnight at 4°C with the primary antibody (mouse monoclonal anti-IFB-2 antibody MH33, 1:1000 dilution in Roti-Block, Developmental Studies Hybridoma Bank, AB_528311, *Francis and Waterston, 1991*). The membrane was washed three times with TBST (20 mM tris(hydroxymethyl)-aminomethane, 0.15 M NaCl, 0.1% Tween 20 (vol/vol), pH 7.6) and then incubated with the secondary antibody (goat anti-mouse IgG antibodies coupled to horseradish peroxidase from DAKO, Santa Clara, CA, USA, #P0447 at 1:5000 in Roti-Block) for 1 hr at room temperature. Chemiluminescence substrate AceGlow (VWR, Darmstadt, Germany, #730-1510) was detected by Fusion Solo (Vilber Lourmat, Eberhardzell, Germany).

## Fluorescence staining with labeled antibodies

Immunostaining was performed as previously described (*Geisler et al., 2020*). In brief, animals were decapitated in 1× PBS on a poly-l-lysine-coated glass slide followed by mounting a coverslip on top. Excessive liquid was removed before freezing the sample in liquid nitrogen. Permeabilization of the cuticle was achieved by rapid removal of the coverslip using a scalpel. Samples were processed as follows: 10 min in methanol, 20 min in acetone (both steps at −20°C), followed by incubation in a graded ethanol series (5 min each in 90%/60% ethanol at −20°C and 5 min in 30% ethanol at room temperature) and 10 min washing in TBST (TBS buffer [20 mM (wt/vol) tris(hydroxymethyl)-aminomethane pH 7.6, 0.15 M (wt/vol) NaCl]+0.2% Tween 20). Samples were then incubated with primary antibodies dissolved in blocking solution (1% (wt/vol) non-fat milk powder (Roth, Karlsruhe, Germany, # T145.2), 1% (wt/vol) bovine serum albumin and 0.02% (wt/vol) sodium azide) overnight at 4°C. After 10 min washing in TBST samples were incubated with secondary antibodies dissolved in blocking solution for 2 hr at room temperature. After washing for 10 min in TBST samples were embedded in Mowiol 4-88 (Sigma-Aldrich, Hamburg, Germany, #81381) supplemented with DABCO (Roth, Karlsruhe, Germany, # 0718.2) and covered with a glass coverslip.

The following primary antibodies were used: Mouse monoclonal anti-IFB-2 antibody MH33 (1:100, Developmental Studies Hybridoma Bank, AB_528311, *Francis and Waterston, 1991*) and rabbit poly-clonal anti-GFP antibody (1:1000, Invitrogen, Carlsbad, CA, #A-11122). Secondary antibodies were Alexa Fluor 488-conjugated affinity-purified anti-mouse IgG (1:200, Invitrogen, Carlsbad, CA, USA, #A-11029) and Alexa Fluor 555-conjugated highly cross-absorbed anti-mouse IgG (1:200, Invitrogen, Carlsbad, CA, USA, #A-21424).

## Analysis of larval development and progeny production

For the analysis of larval development, a defined number of isolated embryos were placed on an NGM plate with an OP50 bacterial lawn and incubated at 18°C. The number of adult stages was then determined daily, and the remaining larval stages were transferred to a new plate. For further in-depth analysis of the different stages of development, the individual age of each larvae was also determined. The larval arrest rate was calculated from the following quotient: number of animals that died as larvae divided by the total number of animals at the beginning of the experiment. Significance was calculated using the Chi square function of Excel (Microsoft, Redmont, WA, USA). Time of development was calculated based on the time it took to complete development from the embryo (up to 24 cell stage) to the adult stage. The offspring of these animals was determined and is presented as average progeny per individual. The generated data are presented as mean value ± SD. Significance was calculated using the unpaired, two-tailed t-test function of GraphPad Prism 5.01 (GraphPad Software Inc, LaJolla, CA, USA). To test for Gaussian distribution we used the D'Agostino & Pearson test of normality. Three independent experiments were performed. Number of n states the number of biological replicates. Sample size was not pre-determined using statistical analysis. Experiments were not randomized and investigators were not blinded during experiments and result analysis.

## Life span analysis

Embryos were isolated, transferred to NGM plates with OP50 bacterial lawns, and then incubated at 18°C. The hatched animals were checked daily for vitality. Animals without reaction to mechanical stimulus, triggered by a platinum wire, were considered dead. Animals that could not be found were censored. In order to avoid mix up with the following generation, the animals were transferred to new plates at least every 3 days. Statistical analysis was performed using the survival function and the Gehan-Breslow-Wilcoxon test of GraphPad Prism 5.01. Three independent experiments were performed. Number of n states the number of biological replicates. Experiments were not randomized and investigators were not blinded during experiments and result analysis.

## Stress assays

For the oxidative stress assay, NGM agar plates containing 200 mM methyl viologen dichloride hydrate (paraquat; Sigma-Aldrich, #856177) were prepared, stored at room temperature overnight, and then inoculated with 50 µl of a 10× concentrated OP50 overnight culture. After another day at room temperature plates were stored at 4°C and were used for a maximum of 3 days. L4 larvae were placed on the bacterial lawn and incubated at 18°C. Every hour the viability of the animals

was checked by mechanically provoking them with a platinum wire. Vital animals showed an active response to this stimulus. Animals that did not respond were considered dead. In parallel, plates without paraquat were used as a control. For statistical analysis, the Gehan-Breslow-Wilcoxon test of GraphPad Prism 5.01 was used.

To perform the osmotic stress assay, plates containing 300 mM NaCl were prepared, stored overnight at room temperature, then inoculated with 300 µl of an OP50 overnight culture and incubated for another day at room temperature. Afterward, L4 stages were placed on the bacterial lawn and plates were incubated overnight at 18°C. The animals were then washed in recovery buffer (M9 buffer with 150 mM NaCl) and transferred to normal NGM plates. After a further overnight incubation at 18°C, single worms were tested for viability as described above. Data shown correspond to mean value ± SD. Statistical calculations were performed using the Chi square function of Excel. Three independent experiments were performed. Number of n states the number of biological replicates. Experiments were not randomized and investigators were not blinded during experiments and result analysis.

## RNAi and body length determination

RNAi by feeding was performed as described previously (*Geisler et al., 2016*) without supplement of tetracycline. In brief, RNAi plates were inoculated with 300 µl of overnight grown HT115 bacteria producing dsRNA targeted against *ifb-2*, *ifc-1/–2*, *ifd-1/–2*, and *ifp-1* mRNA followed by overnight incubation at room temperature. L4 larvae were placed onto plates and incubated for 48 hr at 18°C. Subsequently, adult grown animals were transferred to new plates and incubated for 4 days at 18°C. Laid progeny was used for imaging and body length measurements. Animals were either imaged on the plate using a Nikon AZ100M stereoscope (Düsseldorf, Germany) equipped with a SONY Alpha 7R (Berlin, Germany) camera or mounted on agar slides using 10% levamisole as anesthetic. Body length was determined using the segmented line tool in combination with the measurement function of Fiji (https://imagej.net/Fiji). Results are shown as mean value ± SD. Significance was calculated using the unpaired, two-tailed t-test function of GraphPad Prism 5.01. To test for Gaussian distribution we used the D'Agostino & Pearson test of normality. Three independent experiments were performed. Number of n states the number of biological replicates. Sample size was not pre-determined using statistical analysis. Experiments were not randomized and investigators were not blinded during experiments and result analysis.

RNAi-inducing bacteria were commercially available through the Vidal library (clone *ifb-2*, *ifc-2*, *ifd-2*, Source BioScience, Nottingham, UK), the Ahringer feeding library (clone *ifc-1*, Source BioScience, Nottingham, UK). RNAi clones for *ifd-1* and *ifp-1* were generated by subcloning 1 kb of the corresponding cDNA into a modified L4440 RNAi feeding vector, containing a linker with *AscI* and *NotI* restriction sites, analogous to the site of insertion. The following primers were used: oSMR57 (aggc gcgccTGACCACCATAGCCGAACTT), oSMR58 (agcggccgcTTTGAAGCCACCAACGTCTG), oSMR59 (aggcgcgccTCAAAACCGGGTTCTCGAGA), oSMR60 (agcggccgcTTCACTGCGGAGGTTGATCT). Vector identity was verified by DNA sequencing in each instance.

## Acknowledgements

We thank Barbara Bonn, Stephanie Brosig, Philipp Kolodziej, Janis Moeller, and Sabine Eisner for their commitment and excellent technical support and Adam Breitscheidel for thoughtful figure arrangement. We are particularly thankful to Christine Richardson and Martin Goldberg (Durham University, UK) for high-pressure freeze electron microscopy. Strains N2 and FK312 were obtained from the Caenorhabditis Genetics Centre (University of Minnesota, Minneapolis, MN, USA). The MH33 monoclonal antibody was obtained from the Developmental Studies Hybridoma Bank developed under the auspices of the NICHD and maintained by the University of Iowa, Department of Biology, Iowa City, IA. Funding: The work was supported by the German Research Council (LE566/14-1, 3) grant to RE Leube, the START Program of the Medical Faculty of RWTH Aachen University to F Geisler (131/20), by the Netherlands Organization for Scientific Research (NWO)-VICI 016.VICI.170.165 grant to M Boxem and the 'Deutsche Forschunggemeinschaft' (DFG, German Research Foundation) grant to V Jankowski by the Transregional Collaborative Research Centre (TRR 219; Project-ID 322900939, subproject S-03), INST 948/4S-1, CRU 5011 project number 445703531, and IZKF Multiorgan complexity in Friedreich Ataxia.

## Additional information

### Funding

| Funder | Grant reference number | Author |
|---|---|---|
| Deutsche Forschungsgemeinschaft | LE566/14-1 | Rudolf E Leube |
| Nederlandse Organisatie voor Wetenschappelijk Onderzoek | (NWO)-VICI 016. VICI.170.165 | Mike Boxem |
| Deutsche Forschungsgemeinschaft | INST 948/4S-1 | Vera Jankowski |
| Deutsche Forschungsgemeinschaft | LE566/14-3 | Rudolf E Leube |
| Transregional Collaborative Research Centre | TRR 219 | Vera Jankowski |
| Transregional Collaborative Research Centre | 322900939 | Vera Jankowski |
| Transregional Collaborative Research Centre | subproject S-03 | Vera Jankowski |
| Transregional Collaborative Research Centre | INST 948/4S-1 | Vera Jankowski |

The funders had no role in study design, data collection and interpretation, or the decision to submit the work for publication.

### Author contributions

Florian Geisler, Conceptualization, Data curation, Formal analysis, Supervision, Funding acquisition, Validation, Investigation, Visualization, Methodology, Writing - original draft, Writing - review and editing; Sanne Remmelzwaal, Ruben Schmidt, Investigation; Vera Jankowski, Resources, Formal analysis, Validation, Investigation, Methodology; Mike Boxem, Supervision, Funding acquisition, Writing - review and editing; Rudolf E Leube, Conceptualization, Resources, Supervision, Funding acquisition, Writing - original draft, Project administration, Writing - review and editing

### Author ORCIDs

Ruben Schmidt http://orcid.org/0000-0001-9187-5424
Mike Boxem http://orcid.org/0000-0003-3966-4173
Rudolf E Leube http://orcid.org/0000-0002-5519-7379

### Decision letter and Author response

Decision letter https://doi.org/10.7554/eLife.82333.sa1
Author response https://doi.org/10.7554/eLife.82333.sa2

## Additional files

### Supplementary files
- Supplementary file 1. Summary of DNA/RNA-based reagents used for CRISPR-Cas9.
- MDAR checklist

### Data availability

The authors confirm that all relevant data are included in the article.

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
