## [Editor Report]

This is a very detailed and rigorous study using the power of worm genetics and phenotypic characterization to investigate defects in intermediate filament assembly and organization and its effects on tissue mechanobiology, particularly in the intestine. The work also has implications for understanding disease pathologies.

---

## [Decision Letter]

**Decision letter after peer review:**

Thank you for submitting your article "Intermediate filament network perturbation in the *C. elegans* intestine causes systemic toxicity" for consideration by *eLife*. Your article has been reviewed by 3 peer reviewers, and the evaluation has been overseen by a Reviewing Editor (Mohan Balasubramanian) and Jonathan Cooper as the Senior Editor. The following individual involved in the review of your submission has agreed to reveal their identity: John E. Eriksson (Reviewer #1). We apologize for the delay caused due to circumstances beyond our control.

Essential revisions:

*Reviewer #1 (Recommendations for the authors):*

Below are a number of questions for the authors to consider and amend correspondingly when relevant.

1. The title may need some rephrasing. From the provided data in the manuscript, it is not obvious that the observed IF network perturbation causes systemic toxicity, as "systemic" is generally referred to as something that affects the whole body rather than a specific organ or area.

2. In Figure 2B, the authors claimed that sma-5(n678) animal develops slower, 'some of them never reach adulthood.' This statement seems overly vague and needs to be specified. The figure shows that there is a significant number of individuals at L4 stage on day5. On what day may one find adult sma-5 (n678) animals? What is the percentage of the sma-5 animals which finally reached adulthood? In other words, to what extent are those animals 'delayed' in development, and what was the percentage that did not reach adulthood?

3. Another consideration is to what extent the manuscript explores the effects of the direct target phosphosite of SMA-5 kinase. While the phosphorylation profile of the head domain seems to have a key or partial role in the observed network perturbation, as a whole, the phosphorylation section seems to need strengthening, especially as there is rather extensive literature on the roles of phosphorylation in regulation specific IF domains. This literature could be referred to in greater detail and it would be interesting to hear whether the authors think that the kinase interaction they describe resembles some of the kinase interactions observed in vertebrate and mammalian IFs.

4. On the same token, SMA-5 is referred to as a MAPK orthologue but with limited or no information on which MAPK family member it would resemble most. Both the introduction and discussion would benefit from placing this MAPK in its vertebrate context. Would there be any analogous MAPK interaction in vertebrates/mammals?

5. The statement at the end of the first result section: 'Taken together, we can conclude that loss of IFB-2 rescues all major SMA-5 phenotypes to levels observed in ifb-2(kc14). This demonstrates that the alterations of the IF network observed in sma-5(n678) exert a gain-of-toxic function by negatively affecting intestinal and organismal physiology. '

The authors compared developmental stages, lifespan, and oxidative stress response between mutant and rescued animals, which shows that alterations in the IF network affected both organ and animal development. However, there seems to be a leap of faith that this delayed effect on animal development is a direct result of the effects on intestinal physiology, as there could be parallel processes at work. The authors may consider rephrasing the claims so that the statement would better match the conclusion that can be drawn from the experiments.

6. SMA-5 mutants are smaller and develop more slowly. Could it be that SMA-5 is part of a cascade/signaling pathway responsible for cell growth so it's actually activating something else?

7. Is there any evolutionary correlation between IFC2 and IFB2?

8. It seems IFC2 is reinforcing the expression of the deficient phenotype. Is IFC-2 playing any role in determining the role of IFB-2, which itself is causing the lack of endotube?

9. Please mark magnified sections in images of lower magnification. For example, in Figure 4A' mark where in the bigger image the magnified crop-out is from.

10. The endotube and lack thereof in figure 5A-D are a bit hard to see unless one knows what to look for, even with the red/green arrows. Here a magnified crop-out (as used in for example 4A') can be used to make things more obvious

11. The findings where IF silencing does not rescue osmotic stress response are very interesting and also relevant. It would be interesting if the discussion and conclusions could further elaborate on these observations.

12. Were there other processes or organs that were impacted due to IF deletion during the course of the study but were not shown among the findings?

13. The discussion focuses on the relationship between IF network organization and cell and tissue function. Based on the model in figure 7, it seems that an aberrant IF network influences function by impacting a major tissue structure (in this case endotube morphology) but a loss of an IF does not necessarily affect cell and tissue function much unless there is a challenge that reveals a protective or other function of a give IF. In the recent review by Ridge, Eriksson, Pekny & Goldman in Genes & Development (2022), there is an extensive discussion on the balance between protective IF functions vs. IF triggering maladaptive responses, such as fibrosis, extensive gliosis, etc. that may lead to dysregulated tissue repair and tissue destruction. It would be advantageous if the authors in the discussion would consider the balance between protective vs. maladaptive IF responses and whether their observations on a toxic gain of function effect could somehow reflect such a maladaptive response.

*Reviewer #2 (Recommendations for the authors):*

The authors argue that their "findings provide strong evidence for a gain-of-toxic function of the deranged IF networks". The term gain-of-toxic function is not clear and may not be appropriate. Also, it is common sense that perturbation of the IF network will lead to impairment of overall cell structure and function. It has been described decades ago to be essential for cellular structure, mechanical stability, and organization, as well as for external and internal signaling.

Overall, the quality of the written text needs improvements at the level of language and rationale.

– Long convoluted sentences.

– Some parts read as if the reader should be familiar with the genes and previous studies.

– Calling the kc20-derived protein a 'residual fragment' (line 127) appears inappropriate as it lacks only 4 amino acids of the full-length protein.

– The statement "example of unique IF network organization"; followed by "this evolutionarily conserved localization of the intestinal IF network" in the introduction is contradicting.

– Line 70/71: …survival and stress sensitivity with the exception of IFC-2 mutants that are also expressed in the excretory canal…; should be called mutated IFC-2 proteins or similar instead.

– Graphical illustrations like in the graphical abstract could accompany the described phenotypes to help visualize the defects.

– Order of references in the main text is not according to publishing dates.

*Reviewer #3 (Recommendations for the authors):*

There is one modest issue that should be rectified prior to publication. In the results, related to figure 4, in which the authors mutate a set of serine residues in the head region to A or D and find little impact of these substitutions. The authors summarize these results stating, lines 291-4 "while the phosphorylation sites in the head domain of IFB-2 are not necessary for IF network morphogenesis per se, they may affect network properties such as subunit turnover that are not readily apparent from static pictures and may result in reduced network resilience leading to the decreased life span." This is fine, however, the abstract concludes: "Mechanistically, IF network morphogenesis is linked to the phosphorylated IFB-2 aminoterminus." which is not an accurate summary of the results.

Some additional suggestions for the authors:

1) The ifb-2 deletion suppressor suggests that assembly of the intestinal IF network requires, and is potentially nucleated by, IFB-2. In their recent Science Reports paper, they state, "The most likely explanation is that IFB-2 pairs with either of the other intestinal IF polypeptides, which may only have a limited or no capacity to form filaments on their own. This view is supported by overlay assays of recombinant polypeptides. They show that IFB-2 binds to IFC-1, IFC-2, IFD-1 and IFD-2, although homo- and heterophilic binding also occurs among the other polypeptides. Overlay assays, however, do not allow to distinguish between different binding strengths and do not provide direct information on filament-forming capacity."

Given the transgenes described in this paper, it would be important to test whether the assembly of IFC-1, IFC-2a/e, IFD-2, and IFP-1 at the endotube is dependent upon IFB-2. This could easily be tested using RNAi to deplete IFB-2. In addition, the discussion of this paper should mention that IFB-2 is required for the assembly of all intestinal IFs. The text on lines 233-239 alludes to these observations, but the term "master regulator" is a bit vague and if it is required for all the IFs to assemble, it would be preferable to say that directly. Similarly on line 427-8, "Our findings furthermore provide strong in vivo evidence that the rescued sma-5 mutant phenotype is caused by the presence of IFB-2-containing pathological assemblies." Again not only do these assemblies contain IFB-2, their assembly requires IFB-2.

2) It would be helpful to have domain diagrams of the various intestinal IFs, particularly IFB-2. This could highlight the various mutations and the head vs rod domains. This summary diagram could also indicate which filaments are known to pair with which other filaments.

[Editors' note: further revisions were suggested prior to acceptance, as described below.]

Thank you for resubmitting your work entitled "Intermediate filament network perturbation in the *C. elegans* intestine causes systemic dysfunctions" for further consideration by *eLife*. Your revised article has been evaluated by Jonathan Cooper (Senior Editor) and a Reviewing Editor (Mohan Balasubramanian).

The manuscript has been improved but there are some remaining issues that need to be addressed, as outlined below. Note that experiments are not required.

1. Figure 3 H

Y-axis, length is misspelled.

2. Line 313 "In one of the CRISPR-induced phosphodeficient IFB-2 mutants (ifb-2(kc27)) we found a deletion of the region E31-A184, corresponding to the coil 1A and L1 domains, in addition to the expected S2/S5/S7/S16-19>A mutations (see Figure 5)."

Perhaps this could be more accurately phrased as follows "During the course of CRISPR mutagenesis, we isolated an allele ifb-2(kc27) which not only contained the intended S2/S5/S7/S16-19>A mutations (see Figure 5), but also suffered a deletion of the region E31-A184, corresponding to the coil 1A and L1 domains.

3. Line 330, no comma is required after "To find out".

4. Line 334ff These data indicate that sma-5 is unlikely to directly phosphorylate ifb-2. Does sma-5 regulate the activity or recruitment of a phosphatase?

5. The paragraph starting at line 502 "By focusing on IF phosphorylation, we obtained evidence that it is involved in perturbed network formation in the sma-5(n678) background." is unclear.

Presumably, the subject of this sentence (the 'it') is IF phosphorylation.

This is not clear nor a rigorous interpretation of the data. While sma-5 mutants certainly impact IF assembly, whether it does so directly or indirectly is not clear. sma-5 mutants do affect IF phosphorylation, but they induce hyperphosphorylation, not hypophosphorylation as one would expect if SMA-5 phosphorylated IFB-2. Nor do the phosphomimic or mimetics phenocopy sma-5 mutants. SMA-5 may regulate IF assembly in a highly indirect manner.

The final paragraph is simultaneously overly vague and too specific. It is overly vague as the mechanism of suppression is very unclear, but this paragraph fails to state this explicitly. It is also overly specific, as the available data do not prove that SMA-5 modulates IF assembly by directly phosphorylating the IF proteins. For example, the data are equally consistent with a model in which SMA-5 phosphorylates a non-IF factor that enables this factor to promote IF assembly and IF assembly could limit IFB-2 phosphorylation. A minor revision stating that the authors made IFB-2 mutants to test whether the IFB-2 is a likely key target of SMA-5, but the results suggest that this is not the case. Further work is needed to understand both the relevant targets of SMA-5 and the role of IF phosphorylation.

---

## [Author Response]

Essential revisions:Reviewer #1 (Recommendations for the authors):Below are a number of questions for the authors to consider and amend correspondingly when relevant.1. The title may need some rephrasing. From the provided data in the manuscript, it is not obvious that the observed IF network perturbation causes systemic toxicity, as "systemic" is generally referred to as something that affects the whole body rather than a specific organ or area.

We consider growth and developmental retardation, increased larval arrest, reduced life span, decreased brood size and increased oxidative stress sensitivity as systemic phenotypes, which we link to IF network perturbation that is restricted to the intestine. Simply removing the perturbed IF network in the intestine rescues the systemic dysfunctions to near wild-type levels. The conclusion formulated in the title is therefore, in our view, the main outcome of our study. We agree with the Reviewer, however, that the chosen wording, i.e. linking "systemic" and "toxicity" may be misleading and therefore suggest the following modified title: "Intermediate filament network perturbation in the *C. elegans* intestine causes systemic dysfunctions".

2. In Figure 2B, the authors claimed that sma-5(n678) animal develops slower, 'some of them never reach adulthood.' This statement seems overly vague and needs to be specified. The figure shows that there is a significant number of individuals at L4 stage on day5. On what day may one find adult sma-5 (n678) animals? What is the percentage of the sma-5 animals which finally reached adulthood? In other words, to what extent are those animals 'delayed' in development, and what was the percentage that did not reach adulthood?

Figure 2C shows that 7.1% of *sma-5(n678)* animals never reach adulthood because of larval arrest, all other animals do. They reach adulthood at 6.0±0.2 days (Figure 2A), i.e. more than two days later than N2 (3.7±0.5 days).

3. Another consideration is to what extent the manuscript explores the effects of the direct target phosphosite of SMA-5 kinase. While the phosphorylation profile of the head domain seems to have a key or partial role in the observed network perturbation, as a whole, the phosphorylation section seems to need strengthening, especially as there is rather extensive literature on the roles of phosphorylation in regulation specific IF domains. This literature could be referred to in greater detail and it would be interesting to hear whether the authors think that the kinase interaction they describe resembles some of the kinase interactions observed in vertebrate and mammalian IFs.

In response to the Reviewer we teamed up with Vera Jankowski for analyzing the phosphorylation of IFB-2 in wild-type and *sma-5(n678)* adult animals. The analyses focused on the rod and tail domains identifying five serine residues and three tyrosine residues that are phosphorylated in the mutant animals. None of the tyrosine residues was found to be phosphorylated in the wild type and only three of the serine residues, albeit at lower frequency than in the mutants. The results are depicted in new Figures 5 and Figure 5—figure supplement 1. We have also added a paragraph on the role of phosphorylation in IF regulation in the Discussion.

4. On the same token, SMA-5 is referred to as a MAPK orthologue but with limited or no information on which MAPK family member it would resemble most. Both the introduction and discussion would benefit from placing this MAPK in its vertebrate context. Would there be any analogous MAPK interaction in vertebrates/mammals?

We added the following: "The *C. elegans* WormBase (version WS287; https://wormbase.org) lists MAPK7 (BMK1; ERK4; ERK5; PRKM7) as its closest orthologue in vertebrates. However, MAPK7 activity has not been linked to IFs so far."

5. The statement at the end of the first result section: 'Taken together, we can conclude that loss of IFB-2 rescues all major SMA-5 phenotypes to levels observed in ifb-2(kc14). This demonstrates that the alterations of the IF network observed in sma-5(n678) exert a gain-of-toxic function by negatively affecting intestinal and organismal physiology. 'The authors compared developmental stages, lifespan, and oxidative stress response between mutant and rescued animals, which shows that alterations in the IF network affected both organ and animal development. However, there seems to be a leap of faith that this delayed effect on animal development is a direct result of the effects on intestinal physiology, as there could be parallel processes at work. The authors may consider rephrasing the claims so that the statement would better match the conclusion that can be drawn from the experiments.

We reworded the section in the following way: "Taken together, we can conclude that loss of IFB-2 rescues all major SMA-5 phenotypes, except osmotic stress hypersensitivity, to levels observed in *ifb-2(kc14)*. This demonstrates that intestinal IF network perturbations adversely affects organismal physiology..." Obviously, the link between both still needs to be worked out (for initial experiments, see, e.g. Coch et al., 2020).

6. SMA-5 mutants are smaller and develop more slowly. Could it be that SMA-5 is part of a cascade/signaling pathway responsible for cell growth so it's actually activating something else?

We think that our experiments rule out this possibility since depletion of IFB-2 alone is sufficient to rescue *sma-5* deficiency. Having said this, however, we do not rule out and consider it quite likely that the SMA-5 activity is not exclusively targeted to the regulation of the IF network. Similarly, we cannot rule out that SMA-5 acts through recruitment of non-IF polypeptides to the IF cytoskeleton. We state this in the modified Discussion; "We posit that the intestine-specific MAPK SMA-5 is linked to one or more of these [i.e., pathways that regulate IF dynamics] signaling pathways, which affect the balance between phosphorylation and dephosphorylation of intestinal IFs and possibly other mediators."

7. Is there any evolutionary correlation between IFC2 and IFB2?

We would like to stress that the intestinal IFs in *C. elegans* are not directly related to vertebrate IFs. For the Reviewer’s confidential perusal, we include an unpublished OrthoMCL cluster analysis of the *C. elegans* intestinal IFs.

8. It seems IFC2 is reinforcing the expression of the deficient phenotype. Is IFC-2 playing any role in determining the role of IFB-2, which itself is causing the lack of endotube?

Published experimental evidence strongly suggests that IFB-2 forms heteropolymers with the other intestinal IF polypeptides (Karabinos et al., 2017). Depletion of IFC-2 therefore compromises the ability of IFB-2 to form filaments and consequently prevents proper endotube formation.

9. Please mark magnified sections in images of lower magnification. For example, in Figure 4A' mark where in the bigger image the magnified crop-out is from.

We made the requested changes.

10. The endotube and lack thereof in figure 5A-D are a bit hard to see unless one knows what to look for, even with the red/green arrows. Here a magnified crop-out (as used in for example 4A') can be used to make things more obvious

We improved the labeling and added the requested micrographs with corresponding explanatory schemes in new Figure 6A-D´´. This resulted in separating former Figure 5 E-I into new Figure 7 A-E. We also added explanatory schemes to Figure 1.

11. The findings where IF silencing does not rescue osmotic stress response are very interesting and also relevant. It would be interesting if the discussion and conclusions could further elaborate on these observations.

Our interpretation of this finding is that osmotic stress resilience relies on the presence of an intact IF network reflecting a fundamental stress-protective function crucial of epithelial barrier formation. We mention this in the Discussion and refer to the pioneering work of Brigitte Lane. "…it has been suggested that osmotic stress resilience is a fundamental function of cytoplasmic IFs (D'Alessandro et al., 2002, Pekny and Lane, 2007). Osmotic challenges are of particular relevance to the intestine and its exposure to microbial toxins."

12. Were there other processes or organs that were impacted due to IF deletion during the course of the study but were not shown among the findings?

We did not notice additional phenotypes except those reported for IFC-2 in the excretory canal (mentioned in the Introduction). We cannot exclude that other phenotypes may have escaped our attention so far. In fact, we even expect that further fine analyses will reveal alterations in other organs due to indirect effects elicited by the perturbed intestinal physiology.

13. The discussion focuses on the relationship between IF network organization and cell and tissue function. Based on the model in figure 7, it seems that an aberrant IF network influences function by impacting a major tissue structure (in this case endotube morphology) but a loss of an IF does not necessarily affect cell and tissue function much unless there is a challenge that reveals a protective or other function of a give IF. In the recent review by Ridge, Eriksson, Pekny & Goldman in Genes & Development (2022), there is an extensive discussion on the balance between protective IF functions vs. IF triggering maladaptive responses, such as fibrosis, extensive gliosis, etc. that may lead to dysregulated tissue repair and tissue destruction. It would be advantageous if the authors in the discussion would consider the balance between protective vs. maladaptive IF responses and whether their observations on a toxic gain of function effect could somehow reflect such a maladaptive response.

We thank the Reviewer for the insightful comment and included some of his ideas in the revised manuscript version together with a referral to the mentioned recent publication in Genes & Development. Besides quoting the paper in the Introduction, we added the following sentence in the Discussion "The scenario is reminiscent to that described in vertebrate disease paradigms, in which IF depletion has positive effects on disease outcome (review in Ridge et al., 2023)."

Reviewer #2 (Recommendations for the authors):The authors argue that their "findings provide strong evidence for a gain-of-toxic function of the deranged IF networks". The term gain-of-toxic function is not clear and may not be appropriate. Also, it is common sense that perturbation of the IF network will lead to impairment of overall cell structure and function. It has been described decades ago to be essential for cellular structure, mechanical stability, and organization, as well as for external and internal signaling.

We are not aware that the books have been closed on whether and to which degree IF aggregate formation serve to sequester misfolded proteins for the benefit of cell function or are by themselves toxic to cell function. We are aware that the debate has been ongoing for decades and that it is difficult/impossible to unequivocally resolve it. But using *C. elegans* as a model system, we have the opportunity to study in a well-characterized and comparatively simple organism under defined conditions consequences that go beyond the specific cell type or tissue. The observed adverse effects for the entire organism are obvious and we are not aware that a similar straightforward genotype-organismal phenotype relationship has been shown for other IF aggregate-forming diseases. Taken the Reviewers' comments into account, however, we have toned down our statements throughout the manuscript and slightly changed the title from "systemic toxicity" to "systemic dysfunction".

Overall, the quality of the written text needs improvements at the level of language and rationale.– Long convoluted sentences.– Some parts read as if the reader should be familiar with the genes and previous studies.– Calling the kc20-derived protein a 'residual fragment' (line 127) appears inappropriate as it lacks only 4 amino acids of the full-length protein.– The statement "example of unique IF network organization"; followed by "this evolutionarily conserved localization of the intestinal IF network" in the introduction is contradicting.– Line 70/71: …survival and stress sensitivity with the exception of IFC-2 mutants that are also expressed in the excretory canal…; should be called mutated IFC-2 proteins or similar instead.– Graphical illustrations like in the graphical abstract could accompany the described phenotypes to help visualize the defects.– Order of references in the main text is not according to publishing dates.

We thank the Reviewer for careful reading the manuscript. We have done our best to improve language and rationale accordingly. We also clarified the misunderstandings. New graphical explanations were added to Figures 1 and 6.

Reviewer #3 (Recommendations for the authors):There is one modest issue that should be rectified prior to publication. In the results, related to figure 4, in which the authors mutate a set of serine residues in the head region to A or D and find little impact of these substitutions. The authors summarize these results stating, lines 291-4 "while the phosphorylation sites in the head domain of IFB-2 are not necessary for IF network morphogenesis per se, they may affect network properties such as subunit turnover that are not readily apparent from static pictures and may result in reduced network resilience leading to the decreased life span." This is fine, however, the abstract concludes: "Mechanistically, IF network morphogenesis is linked to the phosphorylated IFB-2 aminoterminus." which is not an accurate summary of the results.

We corrected the summary as requested and include the new results: "Mechanistically, perturbed IF network morphogenesis is linked to hyperphosphorylation of multiple sites throughout the entire IFB-2 molecule."

Some additional suggestions for the authors:1) The ifb-2 deletion suppressor suggests that assembly of the intestinal IF network requires, and is potentially nucleated by, IFB-2. In their recent Science Reports paper, they state, "The most likely explanation is that IFB-2 pairs with either of the other intestinal IF polypeptides, which may only have a limited or no capacity to form filaments on their own. This view is supported by overlay assays of recombinant polypeptides. They show that IFB-2 binds to IFC-1, IFC-2, IFD-1 and IFD-2, although homo- and heterophilic binding also occurs among the other polypeptides. Overlay assays, however, do not allow to distinguish between different binding strengths and do not provide direct information on filament-forming capacity."Given the transgenes described in this paper, it would be important to test whether the assembly of IFC-1, IFC-2a/e, IFD-2, and IFP-1 at the endotube is dependent upon IFB-2. This could easily be tested using RNAi to deplete IFB-2. In addition, the discussion of this paper should mention that IFB-2 is required for the assembly of all intestinal IFs. The text on lines 233-239 alludes to these observations, but the term "master regulator" is a bit vague and if it is required for all the IFs to assemble, it would be preferable to say that directly. Similarly on line 427-8, "Our findings furthermore provide strong in vivo evidence that the rescued sma-5 mutant phenotype is caused by the presence of IFB-2-containing pathological assemblies." Again not only do these assemblies contain IFB-2, their assembly requires IFB-2.

We performed the suggested RNAi experiments and present them in new Figure 3—figure supplement 1. We find that IFB-2 is essential for assembly of IFC-2, IFD-1, and IFD-2. A very residual apical localization is still detectable for IFC-1 and IFP-1 in the presence of *ifb-2*(RNAi). Note, however, that IFC-1 and IFP-1 fluorescence was not due to expression of the respective endogenous genes but due to extrachromosomal fosmids, which may have resulted in overexpression.

In light of these data, we now refer to IFB-2 as a "main" regulator that co-polymerizes with most if not all other intestinal IFs.

2) It would be helpful to have domain diagrams of the various intestinal IFs, particularly IFB-2. This could highlight the various mutations and the head vs rod domains. This summary diagram could also indicate which filaments are known to pair with which other filaments.

A domain diagram is provided in new Figure 5 outlining the phosphorylation sites. Given the known ambiguity on IF polypeptide co-polymerization, we abstain from including this aspect in a formal scheme.

[Editors' note: further revisions were suggested prior to acceptance, as described below.]

The manuscript has been improved but there are some remaining issues that need to be addressed, as outlined below. Note that experiments are not required.1. Figure 3 HY-axis, length is misspelled.

Thank you for pointing our mistake, which we corrected.

2. Line 313 "In one of the CRISPR-induced phosphodeficient IFB-2 mutants (ifb-2(kc27)) we found a deletion of the region E31-A184, corresponding to the coil 1A and L1 domains, in addition to the expected S2/S5/S7/S16-19>A mutations (see Figure 5)."Perhaps this could be more accurately phrased as follows "During the course of CRISPR mutagenesis, we isolated an allele ifb-2(kc27) which not only contained the intended S2/S5/S7/S16-19>A mutations (see Figure 5), but also suffered a deletion of the region E31-A184, corresponding to the coil 1A and L1 domains.

Thank you very much for the thoughtful suggestion which we have adopted in the corrected manuscript version as follows:

"During the course of CRISPR mutagenesis, we isolated allele *ifb-2(kc27)*. This allele codes for an IFB-2 mutant containing not only the intended S2/S5/S7/S16-19>A mutations (see Figure 5) but a deletion of the region E31-A184, corresponding to the coil 1A and L1 domains".

3. Line 330, no comma is required after "To find out".

Corrected.

4. Line 334ff These data indicate that sma-5 is unlikely to directly phosphorylate ifb-2. Does sma-5 regulate the activity or recruitment of a phosphatase?

Thanks for the suggestion, which we incorporated by adding the following statement at the end of the paragraph:

"They" [Our observations] "also indicate that IFB-2 is unlikely a direct target of SMA-5. Further experiments are needed to unravel how SMA-5 acts on IFB-2 phosphorylation, which may be accomplished by regulating the activity or recruitment of a phosphatase or by an even more indirect mechanism."

5. The paragraph starting at line 502 "By focusing on IF phosphorylation, we obtained evidence that it is involved in perturbed network formation in the sma-5(n678) background." is unclear.Presumably, the subject of this sentence (the 'it') is IF phosphorylation.This is not clear nor a rigorous interpretation of the data. While sma-5 mutants certainly impact IF assembly, whether it does so directly or indirectly is not clear. sma-5 mutants do affect IF phosphorylation, but they induce hyperphosphorylation, not hypophosphorylation as one would expect if SMA-5 phosphorylated IFB-2. Nor do the phosphomimic or mimetics phenocopy sma-5 mutants. SMA-5 may regulate IF assembly in a highly indirect manner.The final paragraph is simultaneously overly vague and too specific. It is overly vague as the mechanism of suppression is very unclear, but this paragraph fails to state this explicitly. It is also overly specific, as the available data do not prove that SMA-5 modulates IF assembly by directly phosphorylating the IF proteins. For example, the data are equally consistent with a model in which SMA-5 phosphorylates a non-IF factor that enables this factor to promote IF assembly and IF assembly could limit IFB-2 phosphorylation. A minor revision stating that the authors made IFB-2 mutants to test whether the IFB-2 is a likely key target of SMA-5, but the results suggest that this is not the case. Further work is needed to understand both the relevant targets of SMA-5 and the role of IF phosphorylation.

We completely agree that multiple mechanisms may be at work linking SMA-5 activity and IFB-2 phosphorylation. Taken the helpful suggestions, we have slightly rewritten the last two paragraphs of the Discussion accordingly.